# LARGE LANGUAGE MODELS AS OPTIMIZERS

**Chengrun Yang**[*]   **Xuezhi Wang**   **Yifeng Lu**   **Hanxiao Liu**
**Quoc V. Le**   **Denny Zhou**   **Xinyun Chen**[*]

```
{chengrun, xuezhiw, yifenglu, hanxiao}@google.com
{qvl, dennyzhou, xinyunchen}@google.com
```

Google DeepMind   [*] Equal contribution

## ABSTRACT

Optimization is ubiquitous. While derivative-based algorithms have been powerful tools for various problems, the absence of gradient imposes challenges on many real-world applications. In this work, we propose Optimization by PROmpting (OPRO), a simple and effective approach to leverage large language models (LLMs) as optimizers, where the optimization task is described in natural language. In each optimization step, the LLM generates new solutions from the prompt that contains previously generated solutions with their values, then the new solutions are evaluated and added to the prompt for the next optimization step. We first showcase OPRO on linear regression and traveling salesman problems, then move on to our main application in prompt optimization, where the goal is to find instructions that maximize the task accuracy. With a variety of LLMs, we demonstrate that the best prompts optimized by OPRO outperform human-designed prompts by up to $8\%$ on GSM8K, and by up to $50\%$ on Big-Bench Hard tasks. Code at `https://github.com/google-deepmind/opro`.

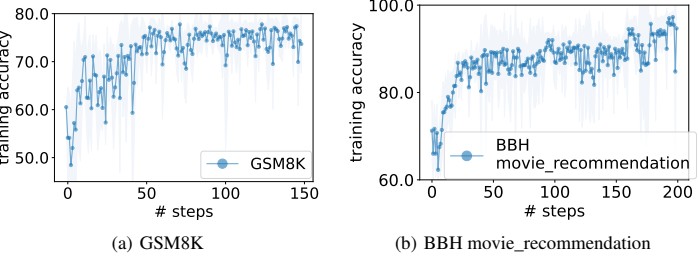

|  (a) GSM8K | (b) BBH movie_recommendation |
|---|---|

Figure 1: Prompt optimization on GSM8K (Cobbe et al., 2021) and BBH (Suzgun et al., 2022) movie_recommendation. The optimization on GSM8K has pre-trained `PaLM 2-L` as the scorer and the instruction-tuned `PaLM 2-L` (denoted `PaLM 2-L-IT`) as the optimizer; the optimization on BBH movie_recommendation has `text-bison` as the scorer and `PaLM 2-L-IT` as the optimizer. Each dot is the average accuracy across all (up to 8) generated instructions in the single step, and the shaded region represents standard deviation. See Section 5 for more details on experimental setup.

Table 1: Top instructions with the highest GSM8K zero-shot test accuracies from prompt optimization with different optimizer LLMs. All results use the pre-trained `PaLM 2-L` as the scorer.

| Source | Instruction | Acc |
|---|---|---|
| *Baselines* | | |
| (Kojima et al., 2022) | Let's think step by step. | 71.8 |
| (Zhou et al., 2022b) | Let's work this out in a step by step way to be sure we have the right answer. | 58.8 |
|  | (empty string) | 34.0 |
| *Ours* | | |
| `PaLM 2-L-IT` | Take a deep breath and work on this problem step-by-step. | **80.2** |
| `PaLM 2-L` | Break this down. | 79.9 |
| `gpt-3.5-turbo` | A little bit of arithmetic and a logical approach will help us quickly arrive at the solution to this problem. | 78.5 |
| `gpt-4` | Let's combine our numerical command and clear thinking to quickly and accurately decipher the answer. | 74.5 |

# 1 INTRODUCTION

Optimization is critical for all areas. Many optimization techniques are iterative: the optimization starts from an initial solution, then iteratively updates the solution to optimize the objective function (Amari, 1993; Qian, 1999; Kingma & Ba, 2015; Bäck & Schwefel, 1993; Rios & Sahinidis, 2013; Reeves, 1993). The optimization algorithm typically needs to be customized for an individual task to deal with the specific challenges posed by the decision space and the performance landscape, especially for derivative-free optimization.

In this work, we propose Optimization by PROmpting (OPRO), a simple and effective approach to utilize large language models (LLMs) as optimizers. With the advancement of prompting techniques, LLMs have achieved impressive performance in various domains (Wei et al., 2022; Kojima et al., 2022; Wang et al., 2022; Zhou et al., 2022a; Madaan et al., 2023; Bai et al., 2022; Chen et al., 2023e). Their ability to understand natural language lays out a new possibility for optimization: instead of formally defining the optimization problem and deriving the update step with a programmed solver, we describe the optimization problem in natural language, then instruct the LLM to iteratively generate new solutions based on the problem description and the previously found solutions. Optimization with LLMs enables quick adaptation to different tasks by changing the problem description in the prompt, and the optimization process can be customized by adding instructions to specify the desired properties of the solutions.

To demonstrate the potential of LLMs for optimization, we first present case studies on linear regression and the traveling salesman problem, which are two classic optimization problems that underpin many others in mathematical optimization, computer science, and operations research. On small-scale optimization problems, we show that LLMs are able to find good-quality solutions simply through prompting, and sometimes match or surpass hand-designed heuristic algorithms.

Next, we demonstrate the ability of LLMs to optimize prompts: the goal is to find a prompt that maximizes the task accuracy. Specifically, we focus on natural language tasks where both the task input and output are texts. LLMs are shown to be sensitive to the prompt format (Zhao et al., 2021; Lu et al., 2021; Wei et al., 2023; Madaan & Yazdanbakhsh, 2022); in particular, semantically similar prompts may have drastically different performance (Kojima et al., 2022; Zhou et al., 2022b; Zhang et al., 2023), and the optimal prompt formats can be model-specific and task-specific (Ma et al., 2023; Chen et al., 2023c). Therefore, prompt engineering is often important for LLMs to achieve good performance (Reynolds & McDonell, 2021). However, the large and discrete prompt space makes it challenging for optimization, especially when only API access to the LLM is available. Following prior work on continuous and discrete prompt optimization (Lester et al., 2021; Li & Liang, 2021; Zhou et al., 2022b; Pryzant et al., 2023), we assume a training set is available to compute the training accuracy as the objective value for optimization, and we show in experiments that optimizing the prompt for accuracy on a small training set is sufficient to reach high performance on the test set.

The prompt to the LLM serves as a call to the optimizer, and we name it the *meta-prompt*. Figure 3 shows an example. The meta-prompt contains two core pieces of information. The first piece is previously generated prompts with their corresponding training accuracies. The second piece is the optimization problem description, which includes several exemplars randomly selected from the training set to exemplify the task of interest. We also provide instructions for the LLM to understand the relationships among different parts and the desired output format. Different from recent work on using LLMs for automatic prompt generation (Zhou et al., 2022b; Pryzant et al., 2023), each optimization step in our work *generates* new prompts that aim to increase the test accuracy based on a trajectory of previously generated prompts, instead of *editing* one input prompt according to natural language feedback (Pryzant et al., 2023) or requiring the new prompt to follow the same semantic meaning (Zhou et al., 2022b). Making use of the full optimization trajectory, OPRO enables the LLM to gradually generate new prompts that improve the task accuracy throughout the optimization process, where the initial prompts have low task accuracies.

We conduct comprehensive evaluation on several LLMs, including `text-bison` and `Palm 2-L` in the PaLM-2 model family (Anil et al., 2023), as well as `gpt-3.5-turbo` and `gpt-4` in the GPT model family. We optimize prompts on GSM8K (Cobbe et al., 2021) and Big-Bench Hard (Suzgun et al., 2022), which are reasoning benchmarks where prompting techniques have achieved remarkable performance breakthrough (Wei et al., 2022; Kojima et al., 2022; Suzgun et al., 2022). Starting from initial prompts with low task accuracies, we show that all LLMs in our evaluation are able to

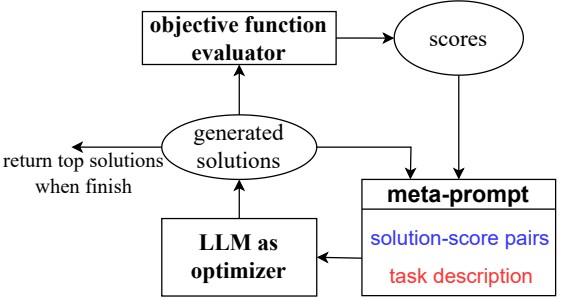

Figure 2: An overview of the OPRO framework. Given the meta-prompt as the input, the LLM generates new solutions to the objective function, then the new solutions and their scores are added into the meta-prompt for the next optimization step. The meta-prompt contains the solution-score pairs obtained throughout optimization, a natural language description of the task, and (in prompt optimization) a few task exemplars. Figure 3 shows a sample meta-prompt for prompt optimization.

serve as optimizers, which consistently improve the performance of the generated prompts through iterative optimization until convergence (see Figure 1). In particular, while these LLMs generally produce instructions of different styles (see Table 1), with zero-shot prompting, their best generated instructions match the few-shot chain-of-thought prompting performance when applied to `PaLM 2-L`, outperforming the zero-shot performance with human-designed prompts by up to $8\%$ on GSM8K. Additionally, we observe that the OPRO-optimized prompts transfer to other benchmarks of the same domain and also deliver notable performance gain.

## 2 OPRO: LLM AS THE OPTIMIZER

Figure 2 illustrates the overall framework of OPRO. In each optimization step, the LLM generates candidate solutions to the optimization task based on the optimization problem description and previously evaluated solutions in the meta-prompt. Then the new solutions are evaluated and added to the meta-prompt for the subsequent optimization process. The optimization process terminates when the LLM is unable to propose new solutions with better optimization scores, or a maximum number of optimization steps has reached. We first outline the desired features of LLMs for optimization, then describe the key design choices based on these desirables.

### 2.1 DESIRABLES OF OPTIMIZATION BY LLMS

**Making use of natural language descriptions.** The main advantage of LLMs for optimization is their ability of understanding natural language, which allows people to describe their optimization tasks without formal specifications. For instance, in prompt optimization where the goal is to find a prompt that optimizes the task accuracy, the task can be described with a high-level text summary along with input-output examples.

**Trading off exploration and exploitation.** The exploration-exploitation trade-off is a fundamental challenge in optimization, and it is important for LLMs serving as optimizers to balance these two competing goals. This means that the LLM should be able to exploit promising areas of the search space where good solutions are already found, while also exploring new regions of the search space so as to not miss potentially better solutions.

### 2.2 META-PROMPT DESIGN

As the input to the optimizer LLM, the meta-prompt contains the following two essential parts.

**Optimization problem description.** The first part is the text description of the optimization problem, including the objective function and solution constraints. For example, for prompt optimization, the LLM can be instructed to "generate a new instruction that achieves a higher accuracy", and we denote such instructions in the meta-prompt as *meta-instructions*. We can also provide customized

meta-instructions as an informal regularization of the generated solutions, such as "the instruction should be concise and generally applicable".

**Optimization trajectory.** Besides understanding natural language instructions, LLMs are also shown to be able to recognize patterns from in-context demonstrations (Wei et al., 2023; Madaan & Yazdanbakhsh, 2022; Mirchandani et al., 2023). Our meta-prompt makes use of this property and instructs the LLM to leverage the optimization trajectory for generating new solutions. Specifically, the optimization trajectory includes past solutions and their optimization scores, sorted in the ascending order. Including optimization trajectory in the meta-prompt allows the LLM to identify similarities of solutions with high scores, encouraging the LLM to build upon existing good solutions to construct potentially better ones without the need of explicitly defining how the solution should be updated.

## 2.3 Solution Generation

At the solution generation step, the LLM generates new solutions with the meta-prompt as input. The following are the key optimization challenges we address in this stage.

**Optimization stability.** In the optimization process, not all solutions achieve high scores and monotonically improve over prior ones. Due to the sensitivity of in-context learning to the prompt, LLM output can be drastically affected by low-quality solutions in the input optimization trajectory, especially at the beginning when the solution space has not been adequately explored. This sometimes results in optimization instability and large variance. To improve stability, we prompt the LLM to generate multiple solutions at each optimization step, allowing the LLM to simultaneously explore multiple possibilities and quickly discover promising directions to move forward.

**Exploration-exploitation trade-off.** We tune the LLM sampling temperature to balance between exploration and exploitation. A lower temperature encourages the LLM to exploit the solution space around the previously found solutions and make small adaptations, while a high temperature allows the LLM to more aggressively explore solutions that can be notably different.

## 3 Motivating Example: Mathematical Optimization

We first demonstrate the potential of LLMs in serving as optimizers for mathematical optimization. We present a case study on linear regression as an example of continuous optimization, and another case study on the Traveling Salesman Problem (TSP) (Jünger et al., 1995; Gutin & Punnen, 2006) as an example of discrete optimization. We present the numerical results in Table 2 and 3, and defer their experiment setups to Appendix B. On both tasks, we see LLMs properly capture the optimization directions on small-scale problems merely based on the past optimization trajectory provided in the meta-prompt. We also see limitations and failure cases of using LLMs as optimizers, and we defer the discussion to Appendix C.

Table 2: Linear regression by optimizer LLMs: the mean $\pm$ standard deviation of the number of steps and the number of unique $(w, b)$ pairs explored before reaching the global optima. Both $w$ and $b$ start from 5 random starting points in $[10, 20]$. We use temperature 1.0 for all models. We run each setting 5 times. The starting points are the same across optimizer LLMs but are different across 5 runs, and are grouped by: within the starting region, outside and close to the starting region, and outside and farther from the starting region. Bold numbers indicate the best among three LLMs in each setting. See Appendix B.1 for experiment setup.

| $w_{\text{true}}$ | $b_{\text{true}}$ | number of steps | | | number of unique $(w, b)$ pairs explored | | |
|---|---|---|---|---|---|---|---|
| | | text-bison | gpt-3.5-turbo | gpt-4 | text-bison | gpt-3.5-turbo | gpt-4 |
| 15 | 14 | $5.8 \pm 2.6$ | $7.6 \pm 4.5$ | $\mathbf{4.0} \pm 1.5$ | $40.0 \pm 12.4$ | $36.0 \pm 15.2$ | $\mathbf{17.2} \pm 5.1$ |
| 17 | 17 | $\mathbf{4.0} \pm 1.8$ | $12.6 \pm 6.0$ | $6.0 \pm 3.7$ | $33.4 \pm 11.7$ | $53.8 \pm 16.9$ | $\mathbf{26.0} \pm 10.6$ |
| 16 | 10 | $\mathbf{3.8} \pm 2.2$ | $10.4 \pm 5.4$ | $6.2 \pm 3.1$ | $30.2 \pm 13.4$ | $42.8 \pm 16.3$ | $\mathbf{24.2} \pm 8.2$ |
| 3 | 5 | $\mathbf{9.8} \pm 2.8$ | $10.8 \pm 2.7$ | $12.2 \pm 2.0$ | $55.8 \pm 16.1$ | $39.6 \pm 10.1$ | $\mathbf{33.0} \pm 4.0$ |
| 25 | 23 | $19.6 \pm 11.4$ | $26.4 \pm 18.3$ | $\mathbf{12.2} \pm 3.7$ | $104.0 \pm 52.3$ | $78.6 \pm 26.2$ | $\mathbf{44.2} \pm 8.3$ |
| 2 | 30 | $\mathbf{31.4} \pm 6.3$ | $42.8 \pm 9.7$ | $38.0 \pm 15.9$ | $126.4 \pm 17.7$ | $125.6 \pm 21.7$ | $\mathbf{99.0} \pm 24.6$ |
| 36 | -1 | $\mathbf{35.8} \pm 6.4$ | $45.4 \pm 16.9$ | $50.4 \pm 18.8$ | $174.0 \pm 28.2$ | $142.2 \pm 31.2$ | $\mathbf{116.4} \pm 32.7$ |

Table 3: Results of the Traveling Salesman Problem (TSP) with different number of nodes $n$, where each $n$ contains 5 problems. "# steps" calculates the mean $\pm$ standard error of optimization steps for successful runs that find the optimal solution. "# successes" counts the number of problems that OPRO results in the optimal solution. When no optimal solution is found for any evaluated problem, the corresponding number of steps is N/A. See Appendix B.2 for experiment setup.

| $n$ | optimality gap (%) | | | | | # steps (# successes) | | |
|---|---|---|---|---|---|---|---|---|
| | NN | FI | text-bison | gpt-3.5-turbo | gpt-4 | text-bison | gpt-3.5-turbo | gpt-4 |
| 10 | $13.0 \pm 1.3$ | $3.2 \pm 1.4$ | $\mathbf{0.0} \pm 0.0$ | $\mathbf{0.0} \pm 0.0$ | $\mathbf{0.0} \pm 0.0$ | $40.4 \pm 5.6$ **(5)** | $46.8 \pm 9.3$ **(5)** | $\mathbf{9.6} \pm 3.0$ **(5)** |
| 15 | $9.4 \pm 3.7$ | $1.2 \pm 0.6$ | $4.4 \pm 1.3$ | $1.2 \pm 1.1$ | $\mathbf{0.2} \pm 0.2$ | N/A (0) | $202.0 \pm 41.1$ **(4)** | $\mathbf{58.5} \pm 29.0$ **(4)** |
| 20 | $16.0 \pm 3.9$ | $\mathbf{0.2} \pm 0.1$ | $30.4 \pm 10.6$ | $4.4 \pm 2.5$ | $1.4 \pm 0.6$ | N/A (0) | $438.0 \pm 0.0$ (1) | $\mathbf{195.5} \pm 127.6$ **(2)** |
| 50 | $19.7 \pm 3.1$ | $\mathbf{9.8} \pm 1.5$ | $219.8 \pm 13.7$ | $133.0 \pm 6.8$ | $11.0 \pm 2.6$ | N/A (0) | N/A (0) | N/A (0) |

> I have some texts along with their corresponding scores. The texts are arranged in ascending order based on their scores, where higher scores indicate better quality.
>
> text:
> Let's solve the problem.
> score:
> 63
>
> (... more instructions and scores ...)
>
> The following exemplars show how to apply your text: you replace  in each input with your text, then read the input and give an output. We say your output is wrong if your output is different from the given output, and we say your output is correct if they are the same.
>
> input:
> Q: Alannah, Beatrix, and Queen are preparing for the new school year and have been given books by their parents. Alannah has 20 more books than Beatrix. Queen has 1/5 times more books than Alannah. If Beatrix has 30 books, how many books do the three have together?
> A: 
> output:
> 140
>
> (... more exemplars ...)
>
> Write your new text that is different from the old ones and has a score as high as possible. Write the text in square brackets.

Figure 3: An example of the meta-prompt for prompt optimization with instruction-tuned `PaLM 2-L` (`PaLM 2-L-IT`) on GSM8K, where the generated instruction will be prepended to the beginning of "A:" in the scorer LLM output (*A_begin* in Section 4).  denotes the position where the generated instruction will be added. The blue text contains solution-score pairs; the purple text describes the optimization task and output format; the orange text are meta-instructions. Appendix E.2 presents the full meta-prompts for other optimizer LLMs.

# 4 APPLICATION: PROMPT OPTIMIZATION

Next, we demonstrate the effectiveness of OPRO on prompt optimization, where the objective is to find the prompt that maximizes task accuracy.

We focus on prompt optimization for natural language tasks, where both the input and output are in the text format. The task is represented as a dataset with training and test splits, where the training set is used to calculate the training accuracy as the objective value during the optimization process, and we compute the test accuracy on the test set after the optimization finishes. While traditional optimization often requires a decently large training set, our experiment shows that a small number or fraction of training samples (e.g., 3.5% of the training set for GSM8K, 20% for Big-Bench Hard) is sufficient. The objective function evaluator is an LLM to which the optimized prompt will be applied,

and it can be the same or different from the LLM for optimization. We denote the LLM for objective function evaluation as the *scorer LLM*, and the LLM for optimization as the *optimizer LLM*.

The output of the optimizer LLM is an *instruction*, which is concatenated to the question part of every exemplar and prompts the scorer LLM. We consider the following positions to insert the instruction: *Q_begin* (the instruction is added before the original question), *Q_end* (the instruction is added after the original question), and *A_begin* (the instruction is added to the beginning of the scorer LLM output. This is applicable to pretrained LLMs without instruction tuning, where the prompt is formatted as a sequence of QA pairs). We show examples of these prompting formats in Appendix D.

Figure 3 shows an example of the meta-prompt for prompt optimization on GSM8K (Cobbe et al., 2021). The meta-prompt includes three parts: task examples – a few exemplars from the training set to show how the task looks like and where the generated instructions will be added to; optimization trajectory – the instructions generated in previous optimization steps, along with their scores (only showing the top ones in ascending order); meta-instructions – the instructions to the optimizer LLM that explain the optimization goal and instruct the model how to use the above information.

## 5 PROMPT OPTIMIZATION EXPERIMENTS

We present prompt optimization results in this section. Our experiments demonstrate that OPRO brings a significant performance gain across the board, with different optimizer and scorer LLMs.

### 5.1 EVALUATION SETUP

**Models.** The LLMs we use as optimizer are pre-trained `PaLM 2-L`, instruction-tuned `PaLM 2-L` (denoted `PaLM 2-L-IT`), `text-bison`, `gpt-3.5-turbo`, and `gpt-4`; the LLMs we use as scorer are pre-trained `PaLM 2-L` and `text-bison`. We search for A_begin instructions on pre-trained `PaLM 2-L`, and for Q_begin and Q_end instructions on `text-bison`.

**Benchmarks.** Our primary evaluation benchmarks for prompt optimization are GSM8K (Cobbe et al., 2021) and Big-Bench Hard (BBH) (Suzgun et al., 2022). To examine the transferability of the optimized instructions, we also evaluate the instructions optimized for GSM8K on two other mathematical reasoning datasets, i.e., MultiArith (Roy & Roth, 2016) and AQuA (Ling et al., 2017).

**Implementation details.** We use temperature 0 (greedy decoding) to evaluate generated instructions. We set the default temperature to 1.0 for optimizer LLMs to generate diverse instructions. In each step, we prompt the optimizer LLM with the meta-prompt 8 times to generate 8 instructions. Our meta-prompt contains the best 20 instructions so far and 3 randomly picked exemplars from the training set. The 3 exemplars are independently sampled at each step to reduce the risk of overfitting on the given exemplars. We study the effect of different hyperparameters in ablation studies (Section 5.3).

### 5.2 MAIN RESULTS

Here we show prompt optimization curves on GSM8K and two BBH tasks. The curves on other BBH tasks are deferred to Appendix I, and the tables containing all accuracy numbers are in Appendix J. Appendix L analyzes overfitting in prompt optimization. Appendix M compares the prompt optimization performance of meta-prompts in OPRO and EvoPrompt (Guo et al., 2023).

#### 5.2.1 GSM8K

For GSM8K, we randomly sample 3.5% examples from the training set, and the same subset is used throughout optimization. This balances evaluation cost with generalization performance. After the optimization finishes, we evaluate the found instructions on the entire GSM8K test set.

Figure 1(a) in Section 1 shows the prompt optimization curve with pre-trained `PaLM 2-L` as scorer and `PaLM 2-L-IT` as optimizer, and the initial instruction is "Let's solve the problem" with a (approximated, and same below) training accuracy 60.5. The curve has an overall upward trend with several leaps throughout the optimization process. The curve also shows a decrease of the variance among the accuracies of instructions generated at each step, indicating the optimizer LLM generates

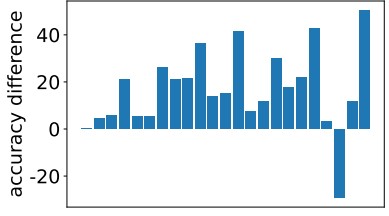
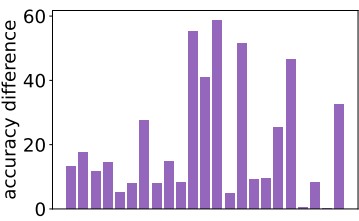

(a) ours minus "Let's think step by step."

(b) ours minus empty starting point

Figure 4: On 23 BBH tasks, the accuracy differences among instructions found by prompt optimization (with the `PaLM 2-L` scorer and the `PaLM 2-L-IT` optimizer), "Let's think step by step.", and the empty string (optimization starting point). The bar charts with task names and those with the `text-bison` scorer are deferred to Figure 19 in Appendix J.1.

*distributionally* better instructions throughout optimization. Appendix F shows similar trends when using `text-bison` as scorer and when using pre-trained `PaLM 2-L` itself as optimizer.

Table 1 summarizes top instructions. We see the styles of instructions found by different optimizer LLMs vary a lot: `PaLM 2-L-IT` and `text-bison` ones are concise; GPT ones are long and detailed. And although some top instructions contain the "step-by-step" phrase, most others achieve a comparable accuracy with different semantic meanings.

### 5.2.2 BBH

On BBH, the optimization starts from an empty string as the initial instruction by default. For each task, we utilize a subset of 20% examples for prompt optimization, and the rest examples are for testing. As shown in Figure 1(b), we also observe an upward trend in the optimization curve on BBH. See Figure 17 and 18 in Appendix I for curves on all BBH tasks from the `text-bison` scorer and the `PaLM 2-L-IT` optimizer. We show experimental results on different scorer and optimizer LLMs, instruction positions, and starting points in Appendix J.

Figure 4 visualizes the per-task accuracy difference on all 23 BBH tasks compared to the instruction "Let's think step by step." (Kojima et al., 2022) and the empty instruction, and we detail the accuracy values in Table 6 of Appendix J. We show that the instructions found by OPRO outperform "Let's think step by step." on almost all tasks by a large margin: with the `PaLM 2-L` scorer, our instructions outperform by over 5% on 19/23 tasks. Our prompt optimization algorithm also improves instructions from the empty starting point by over 5% on most tasks: 20/23 with the `PaLM 2-L` scorer.

### 5.2.3 TRANSFERABILITY OF FOUND INSTRUCTIONS

We assess the transferability of found prompts to different datasets in the same domain, where we evaluate the top instructions found for GSM8K on MultiArith and AQuA. Table 4 shows our optimized prompts also outperform baseline prompts with different scorer LLMs on these two benchmarks.

### 5.3 ABLATION STUDIES

We use `text-bison` as scorer and `PaLM 2-L` as optimizer for all ablation studies. The tasks we evaluate are GSM8K (math reasoning) and BBH sports_understanding (non-math reasoning).

**Meta-prompt design: instruction ordering.** The meta-prompt design is crucial in achieving good prompt optimization performance. Here we investigate the order of the previous instructions. We compare the following: (1) lowest to highest (default setting); (2) highest to lowest; (3) random. Figures 5(a) and 5(b) show that the default setting achieves better final accuracies and converges faster. One hypothesis is that the optimizer LLM output is affected more by the past instructions closer to the end of the meta-prompt. This is consistent with the recency bias observed in Zhao et al. (2021), which states that LLMs are more likely to generate tokens similar to the end of the prompt.

**The number of generated instructions per step.** Computing a mini-batch of gradients reduces the variance of a stochastic gradient descent procedure. Similarly, generating multiple instructions in each

Table 4: Transferability across datasets: accuracies of top instructions found for GSM8K on Multi-Arith and AQuA.

| Scorer | Source | Instruction position | Instruction | Accuracy MultiArith | AQuA |
|---|---|---|---|---|---|
| *Baselines* | | | | | |
| PaLM 2-L | (Kojima et al., 2022) | A_begin | Let's think step by step. | 85.7 | 44.9 |
| PaLM 2-L | (Zhou et al., 2022b) | A_begin | Let's work this out in a step by step way to be sure we have the right answer. | 72.8 | 48.4 |
| PaLM 2-L | | A_begin | Let's solve the problem. | 87.5 | 44.1 |
| PaLM 2-L | | A_begin | (empty string) | 69.3 | 37.8 |
| text-bison | (Kojima et al., 2022) | Q_begin | Let's think step by step. | 92.5 | 31.9 |
| text-bison | (Zhou et al., 2022b) | Q_begin | Let's work this out in a step by step way to be sure we have the right answer. | 93.7 | 32.3 |
| text-bison | | Q_begin | Let's solve the problem. | 85.5 | 29.9 |
| text-bison | | Q_begin | (empty string) | 82.2 | 33.5 |
| *Ours* | | | | | |
| PaLM 2-L | PaLM 2-L-IT on GSM8K | A_begin | Take a deep breath and work on this problem step-by-step. | **95.3** | **54.3** |
| text-bison | PaLM 2-L-IT on GSM8K | Q_begin | Let's work together to solve math word problems! First, we will read and discuss the problem together to make sure we understand it. Then, we will work together to find the solution. I will give you hints and help you work through the problem if you get stuck. | **96.8** | **37.8** |

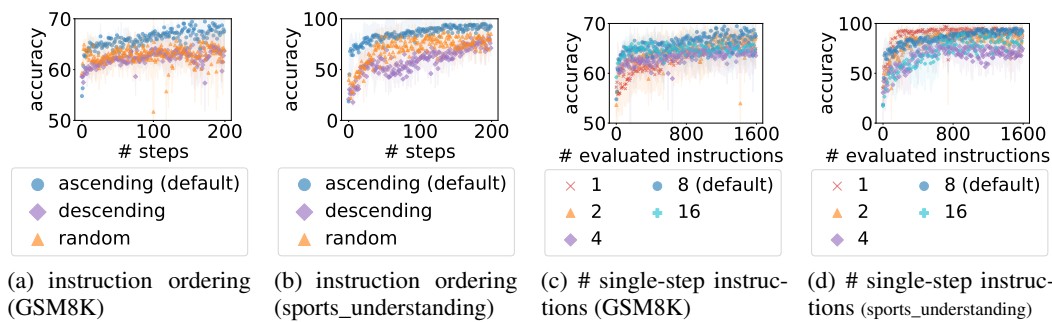

(a) instruction ordering (GSM8K)  (b) instruction ordering (sports_understanding)  (c) # single-step instructions (GSM8K)  (d) # single-step instructions (sports_understanding)

Figure 5: **Ablation studies.** The dots are the average values across 3 optimization repetitions, and the shaded regions represent standard deviations. In Figure (c) and (d), the x-axis represents the total number of evaluated instructions through the optimization; e.g., we run 200 optimization steps when sampling 8 instructions in each step, run 400 steps when sampling 4 instructions in each step, etc.

step improves the optimization stability with LLMs. On the other hand, to achieve better performance with a fixed budget for the number of instructions to evaluate, the number of per-step instructions should not be too large, so as to allow more optimization steps to incorporate richer information of past instructions with their accuracies. Taking both aspects into consideration, Figure 5(c) and 5(d) compares the optimization performance of sampling 1 / 2 / 4 / 8 (default) / 16 instructions in each step, showing that sampling 8 instructions at each step overall achieves the best performance.

We show ablation studies of other algorithmic aspects in Appendix K.

## 6 RELATED WORK

**Prompt optimization.** Prior works have developed soft prompt-tuning methods that optimize the prompt represented as task-specific continuous vectors (Lester et al., 2021; Li & Liang, 2021; Liu et al.,

2021; Qin & Eisner, 2021), as well as performing discrete prompt optimization by gradient-guided search (Shin et al., 2020; Wen et al., 2023; Gao et al., 2020; Chen et al., 2023d) and reinforcement learning (Deng et al., 2022; Zhang et al., 2023). These approaches become inapplicable when there is only API access to the LLM. Other works designed edit-based approaches for gradient-free prompt optimization (Xu et al., 2022; Prasad et al., 2022), where the editing can be done with human-defined operations (e.g., swapping two phrases) (Prasad et al., 2022) or language models (e.g., back translation) (Xu et al., 2022). Some recent works investigate LLMs for prompt optimization (Zhou et al., 2022b; Pryzant et al., 2023; Xu et al., 2023). Specifically, APE (Zhou et al., 2022b) first uses the LLM to generate initial instructions. Afterwards, APE selects top instructions with the highest accuracies, then prompts the LLM with each individual instruction to generate a semantically similar variant of the initial instruction. APO (Pryzant et al., 2023) in each step instructs the LLM to produce text feedback on how to update an old instruction. Different from edit-based approaches, the optimizer LLM in our work directly generates new instructions at each optimization step, and the optimizer LLM is merely asked to improve the task accuracy without being required to imitate past instructions. Compared to Zhou et al. (2022b) and Pryzant et al. (2023), our optimization process incorporates the past generated instructions with their scores in the meta-prompt, enabling the optimizer LLM to discover common patterns of high-quality instructions.

**Tuning language models for optimization.** Some previous works tune or prompt language models to behave as mutation and crossover operators in evolutionary algorithms. Meyerson et al. (2023) uses language models with few-shot exemplars to propose evolutionary cross-overs on tasks like image and code generation. In Lehman et al. (2022), the LLM trained on code diff generation is used as the mutation operator, and they further design a fine-tuning method to improve performance in the Sodarace domain for robot simulation. EvoPrompting (Chen et al., 2023a) uses LLMs to evolve neural network architectures, where they combine evolution with soft prompt tuning. With respect to taking the trajectory as the input for optimization, OptFormer (Chen et al., 2022) trains a transformer model on large collections of hyperparameter optimization data. On the other hand, our work performs optimization solely by prompting without additional training.

## 7 CONCLUSION

We embark on employing LLMs as optimizers, where the LLM progressively generates new solutions to optimize an objective function. We first motivate OPRO with linear regression and traveling salesman problems, then proceed to prompt optimization as a concrete application. Our evaluation demonstrates that LLMs have the capacity of gradually improving the generated solutions based on the past optimization trajectory. Interestingly, on small-scale traveling salesman problems, OPRO performs on par with some hand-crafted heuristics. For prompt optimization, optimized prompts outperform human-designed on GSM8K and BBH by a significant margin, sometimes over $50\%$.

A number of unresolved questions are open for future research on LLMs for optimization. In general, how to reduce the sensitivity to initialization and better balance exploitation with exploration remains a challenge. Specifically, for prompt optimization, one limitation of our current implementation is that the optimizer LLM does not effectively utilize error cases in the training set to infer promising directions to improve the generated instructions. In our experiments, we tried including error cases in the meta-prompt rather than randomly sampling from the training set at each optimization step, but the results are similar, indicating that the error cases alone are not informative enough for the optimizer LLM to grasp the cause of the wrong prediction. Another limitation is that prompt optimization requires a training set to compute the accuracy that guides the optimization process. Currently the training set at least contains tens of samples, so that the optimized prompt does not severely overfit to the training samples. A promising direction is to incorporate richer feedback about the error cases besides the aggregated accuracy, and summarize the key features that distinguish between high-quality and low-quality generated prompts in the optimization trajectory. Such information may inform the optimizer LLM of how to more efficiently improve over the past generated instructions, and potentially further reduce the example set size needed for prompt optimization.

ETHICS STATEMENT

This work uses synthetic math problems for linear regression and traveling salesman problems, and uses public datasets like GSM8K and Big-Bench Hard for prompt optimization. These tasks have been commonly used in similar works and should not be regarded controversial. There is a peril that LLMs may generate harmful information that poses safety risks; how to safeguard model behavior remains valuable future work.

REPRODUCIBILITY STATEMENT

We evaluate on public benchmarks. The `text-bison` API is available at: `https://cloud.google.com/vertex-ai/docs/generative-ai/learn/models`. The GPT models are available here: `http://openai.com/api/`. This work uses `gpt-3.5-turbo-0613` and `gpt-4-0613`.

ACKNOWLEDGMENTS

We thank Daiyi Peng, Yanqi Zhou, Jerry Wei, Shuo Chen, Tim Rocktäschel, Chrisantha Fernando, Dylan Banarse, Henryk Michalewski, Simon Osindero, and Ed H. Chi for their valuable feedback, and thank several anonymous reviewers for helpful comments.

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

## A  RELATED WORK ON PROMPTING WITH NATURAL LANGUAGE FEEDBACK

A recent line of work investigates approaches to improve the LLM performance by prompting with natural language feedback to revise the model output, which has shown effectiveness in reducing harmful LLM outputs (Bai et al., 2022; Ganguli et al., 2023), improving reasoning (Shinn et al., 2023; Madaan et al., 2023) and code generation performance (Chen et al., 2023e; Olausson et al., 2023; Shinn et al., 2023; Chen et al., 2023b), dialogue applications (Nair et al., 2023; Madaan et al., 2023; Yuan et al., 2023), and so on (Kim et al., 2023; Wang et al., 2023). Specifically, Yuan et al. (2023) develops a human-in-the-loop framework for deriving system-level feedback from a collection of instance-level feedback, which is then used for refining data. In our work, the optimizer LLM utilizes the optimization trajectory in the prompt, which implicitly requires the LLM to summarize the common characteristics among solutions with similar scores. We consider incorporating explicit natural language feedback on generated solutions for later optimization steps as future work.

## B  EXPERIMENT SETUPS OF MOTIVATING EXAMPLES ON MATHEMATICAL OPTIMIZATION

### B.1  LINEAR REGRESSION

In linear regression problems, the goal is to find the linear coefficients that probabilistically best explain the response from the input variables. We study the setting in which the independent and dependent variables $X$ and $y$ are both one-dimensional and an intercept $b$ is present, so that there are two one-dimensional variables $w$, $b$ to optimize over. In a synthetic setting, we sample ground truth values for one-dimensional variables $w_{\text{true}}$ and $b_{\text{true}}$, and generate 50 data points by $y = w_{\text{true}}x + b_{\text{true}} + \epsilon$, in which $x$ ranges from 1 to 50 and $\epsilon$ is the standard Gaussian noise. Our optimization starts from 5 randomly sampled $(w, b)$ pairs. In each step, we prompt an instruction-tuned LLM with a meta-prompt that includes the best 20 $(w, b)$ pairs in history and their sorted objective values. The meta-prompt then asks for a new $(w, b)$ pair that further decreases the objective value. A sample meta-prompt is shown in Figure 12 of Appendix E.1. We prompt the meta-prompt 8 times to generate at most 8 new $(w, b)$ pairs in each step to improve optimization stability. Then we evaluate the objective value of the proposed pair and add it to history. We do black-box optimization: the analytic form does not appear in the meta-prompt text. This is because the LLM can often calculate the solution directly from the analytic form.

Table 2 in main paper Section 3 summarizes the results with one of the following optimizer LLMs: `text-bison`, `gpt-3.5-turbo`, and `gpt-4`. We study three settings of $w_{\text{true}}$ and $b_{\text{true}}$: within the starting region $[10, 20] \times [10, 20]$, "near outside" (each of $w_{\text{true}}$ and $b_{\text{true}}$ is outside the starting region but the distance is less than 10), and "far outside" (each of $w_{\text{true}}$ and $b_{\text{true}}$ is outside the starting region and the distance is greater than 10). We see:

- The number of unique $(w, b)$ pairs explored by each model is fewer than exhaustive search, indicating these models are able to to do black-box optimization: compare the numbers and propose a descent direction.
- The `text-bison` and `gpt-4` models outperform `gpt-3.5-turbo` in convergence speed: they arrive at the optima with fewer steps. The `gpt-4` model also outperforms in finding the optima with fewer explored unique points. Taking a closer look at the optimization trajectory, we see `gpt-4` is the best at proposing a reasonable next step from the history: for example, when the history shows the objective values of $(w, b) = (8, 7)$, $(w, b) = (8, 6)$, and $(w, b) = (8, 5)$ are decreasing, it has a highest chance to propose $(w, b) = (8, 4)$ for evaluation.
- The problem becomes harder for all models when the ground truth moves farther from the starting region: all models need more explorations and more steps.

### B.2  TRAVELING SALESMAN PROBLEM (TSP)

TSP is a classical combinatorial optimization problem with numerous algorithms proposed in literature, including heuristic algorithms and solvers (Rosenkrantz et al., 1977; Golden et al., 1980; Optimization et al., 2020; Applegate et al., 2006; Helsgaun, 2017), and approaches based on training deep neural networks (Kool et al., 2019; Deudon et al., 2018; Chen & Tian, 2019; Nazari et al., 2018).

Specifically, given a set of $n$ nodes with their coordinates, the TSP task is to find the shortest route that traverses all nodes from the starting node and finally returns to the starting node.

Our optimization process with LLMs starts from 5 randomly generated solutions, and each optimization step produces at most 8 new solutions. We present the meta-prompt in Figure 13 of Appendix E.1. We generate the problem instances by sampling $n$ nodes with both $x$ and $y$ coordinates in $[-100, 100]$. We use the Gurobi solver (Optimization et al., 2020) to construct oracle solutions and compute the optimality gap for all approaches, where the optimality gap is defined as the difference between the distance in the solution constructed by the evaluated approach and the distance achieved by the oracle solution, divided by the distance of the oracle solution. Besides evaluating OPRO with different LLMs including `text-bison`, `gpt-3.5-turbo` and `gpt-4`, we also compare OPRO to the Nearest Neighbor (NN) and Farthest Insertion (FI) heuristics.

We present the results in main paper Section 3 Table 3. We randomly generate 5 problem instances for each number of nodes $n$. In addition to measuring the optimality gap, on problems where the LLM finds the optimal solutions, we also show the number of optimization steps taken to reach the global optimum. First, we observe that `gpt-4` significantly outperforms `gpt-3.5-turbo` and `text-bison` across all problem sizes. Specifically, on smaller-scale problems, `gpt-4` reaches the global optimum about $4\times$ faster than other LLMs. On larger-scale problems, especially with $n = 50$, `gpt-4` still finds solutions with a comparable quality to heuristic algorithms, while both `text-bison` and `gpt-3.5-turbo` get stuck at local optima with up to $20\times$ worse optimality gaps.

On the other hand, the performance of OPRO degrades dramatically on problems with larger sizes. When $n = 10$, all LLMs find the optimal solutions for every problem; as the problem size gets larger, the OPRO optimality gaps increase quickly, and the FI heuristic starts to outperform all LLMs.

## C  SOME FAILURE CASES

Although LLMs show the power of optimizing basic math problems (Section 3) and prompts (Section 4), we see some limitations across all optimizer LLMs that may impede their power of solving more challenging problems. These limitations include:

- **Hallucinating the values that need to come from math calculation**: The optimizer LLMs often output contents like "the function value at (5, 3) is 15" despite that the true value is not 15. The model will get it right if external tools that can reliably calculate the value are triggered. When and how to trigger such tool use cases remains an interesting topic (see e.g., (Schick et al., 2023; Cai et al., 2023)).

- **Generating solutions already appeared in context even if we tell it to "Give me a new (w, b) pair that is different from all pairs above"**: the optimizer LLMs do not 100% reliably follow this instruction even if its own outputs often include sentences like "I will provide a new pair that is different", making the output self-contradictory. The output is almost guaranteed to be different from in-context old solutions when the model output contains a comparison of the new pair and all old pairs, though. Thus (implicitly) triggering such behaviors may be a solution. How to implement this feature without harming the instruction following performance of other parts remains an interesting topic to study.

- **In black-box math optimization, getting stuck at a point that is neither global nor local optimal**: This often occurs in two linear regression cases: (a) The in-context exemplars all share the same $w$ or $b$ that is different from $w_{\text{true}}$ or $b_{\text{true}}$. This case is more likely to be avoided when a larger number of past solutions are included in the meta-prompt; (b) one or several of the best previous solutions in the meta-prompt have $w$s and $b$s in quantitatively opposite directions from the global optima $w_{\text{true}}$ and $b_{\text{true}}$: for example, the $w$s are all smaller than $w_{\text{true}}$ while the $b$s are all larger than $b_{\text{true}}$. Since the optimizer model often proposes to only increase $w$ or decrease $b$ when the past solutions in meta-prompt share $w$ or $b$, the optimization will get stuck if either increasing $w$ or decreasing $b$ would increase the objective value. This issue is mitigated by sampling multiple new solutions (thus more exploration) at each step.

- **Hard to navigate a bumpy loss landscape**: Like other optimizers, it is harder for the optimizer LLM to optimize black-box functions when the loss landscape gets more complicated. For example, when minimizing the Rosenbrock function $f(x, y) = (a-x)^2 + b(y-x^2)^2$ with $a = 20$

(whose global optimal point is $x = 20$, $y = 400$) with 5 starting points in $[10, 20] \times [10, 20]$, the optimization often gets stuck at around $(0, 0)$. This is because the optimizer LLM sees a decrease of objective value when it drastically decreases both $x$ and $y$ to 0. Then starting from $(0, 0)$, the optimizer LLM is hard to further navigate $x$ and $y$ along the narrow valley in the loss landscape towards $(20, 400)$ (Figure 6).

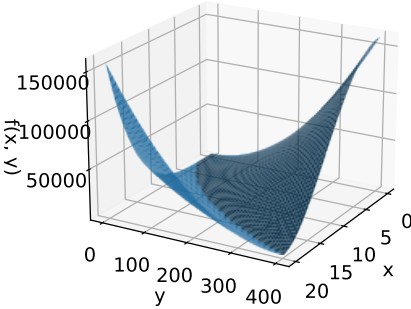

Figure 6: A visualization of the landscape of the Rosenbrock function $f(x, y) = (a - x)^2 + b(y - x^2)^2$ with $a = 20$ and $b = 1$. The global optima is at $x = 20$, $y = 400$ with function value 0. The function value at $x = 0$, $y = 0$ is 400. The landscape has a narrow valley between $(0, 0)$ and $(20, 400)$.

## D    PROMPTING FORMATS FOR SCORER LLM

Figure 7, 8, and 9 show examples of the Q_begin, Q_end, and A_begin prompting formats when the "QA" pattern is present. The "QA" pattern is eliminated when prompting instruction-tuned scorer models like `text-bison` with the Q_begin and Q_end formats (Figure 10 and 11).

---

Q: {instruction}
Janet's ducks lay 16 eggs per day. She eats three for breakfast every morning and bakes muffins for her friends every day with four. She sells the remainder at the farmers' market daily for $2 per fresh duck egg. How much in dollars does she make every day at the farmers' market?

A:

---

Figure 7: The Q_begin prompting format on a GSM8K test exemplar with the "QA" pattern.

---

Q: Janet's ducks lay 16 eggs per day. She eats three for breakfast every morning and bakes muffins for her friends every day with four. She sells the remainder at the farmers' market daily for $2 per fresh duck egg. How much in dollars does she make every day at the farmers' market?
{instruction}

A:

---

Figure 8: The Q_end prompting format on a GSM8K test exemplar with the "QA" pattern.

Q: Janet's ducks lay 16 eggs per day. She eats three for breakfast every morning and bakes muffins for her friends every day with four. She sells the remainder at the farmers' market daily for \$2 per fresh duck egg. How much in dollars does she make every day at the farmers' market?

A: {instruction}

Figure 9: The A_begin prompting format on a GSM8K test exemplar.

{instruction}
Janet's ducks lay 16 eggs per day. She eats three for breakfast every morning and bakes muffins for her friends every day with four. She sells the remainder at the farmers' market daily for \$2 per fresh duck egg. How much in dollars does she make every day at the farmers' market?

Figure 10: The Q_begin prompting format on a GSM8K test exemplar without the "QA" pattern.

Janet's ducks lay 16 eggs per day. She eats three for breakfast every morning and bakes muffins for her friends every day with four. She sells the remainder at the farmers' market daily for \$2 per fresh duck egg. How much in dollars does she make every day at the farmers' market?
{instruction}

Figure 11: The Q_end prompting format on a GSM8K test exemplar without the "QA" pattern.

# E    META-PROMPTS

## E.1    META-PROMPT FOR MATH OPTIMIZATION

Now you will help me minimize a function with two input variables w, b. I have some (w, b) pairs and the function values at those points. The pairs are arranged in descending order based on their function values, where lower values are better.

input:
w=18, b=15
value:
10386334

input:
w=17, b=18
value:
9204724

Give me a new (w, b) pair that is different from all pairs above, and has a function value lower than any of the above. Do not write code. The output must end with a pair [w, b], where w and b are numerical values.

Figure 12: An example of the meta-prompt for linear regression. The blue text contains solution-score pairs; the orange text are meta-instructions.



You are given a list of points with coordinates below: (0): (-4, 5), (1): (17, 76), (2): (-9, 0), (3): (-31, -86), (4): (53, -35), (5): (26, 91), (6): (65, -33), (7): (26, 86), (8): (-13, -70), (9): (13, 79), (10): (-73, -86), (11): (-45, 93), (12): (74, 24), (13): (67, -42), (14): (87, 51), (15): (83, 94), (16): (-7, 52), (17): (-89, 47), (18): (0, -38), (19): (61, 58).
Below are some previous traces and their lengths. The traces are arranged in descending order based on their lengths, where lower values are better.

<trace> 0,13,3,16,19,2,17,5,4,7,18,8,1,9,6,14,11,15,10,12 </trace>
length:
2254

<trace> 0,18,4,11,9,7,14,17,12,15,10,5,19,3,13,16,1,6,8,2 </trace>
length:
2017

<trace> 0,11,4,13,6,10,8,17,12,15,3,5,19,2,1,18,14,7,16,9 </trace>
length:
1953

<trace> 0,10,4,18,6,8,7,16,14,11,2,15,9,1,5,19,13,12,17,3 </trace>
length:
1840

Give me a new trace that is different from all traces above, and has a length lower than any of the above. The trace should traverse all points exactly once. The trace should start with <trace> and end with </trace>.



Figure 13: An example of the meta-prompt for Traveling Salesman Problems with problem size $n = 20$. The blue text contains solution-score pairs; the orange text are meta-instructions.

## E.2 META-PROMPT FOR PROMPT OPTIMIZATION

Different optimizer models work the best on different styles of meta-prompts. Figure 3 in the main paper shows the meta-prompt for `PaLM 2-L-IT`; Figure 14 shows that for pre-trained `PaLM 2-L`; Figure 15 shows that for GPT models.



Create a piece of text at the beginning of the answer to enhance the precision in solving diverse grade school math problems.

Precision: 4 <TEXT>A dime</TEXT>

Precision: 17 <TEXT>The answer is a function. It is</TEXT>

Precision: 19 <TEXT>So how can we find out what this equation means?</TEXT>

Precision: 20 <TEXT>Solutions:</TEXT>



Figure 14: An example of the meta-prompt for prompt optimization with pre-trained `PaLM 2-L` on GSM8K, where the generated instruction will be prepended to the beginning of the scorer LLM output (*A_begin* in Section 4).

---

Your task is to generate the instruction \. Below are some previous instructions with their scores. The score ranges from 0 to 100.

text:
Let's figure it out!
score:
61

text:
Let's solve the problem.
score:
63

(... more instructions and scores ...)

Below are some problems.

Problem:
Q: Alannah, Beatrix, and Queen are preparing for the new school year and have been given books by their parents. Alannah has 20 more books than Beatrix. Queen has 1/5 times more books than Alannah. If Beatrix has 30 books, how many books do the three have together?
A: \

Ground truth answer:
140

(... more exemplars ...)

Generate an instruction that is different from all the instructions \ above, and has a higher score than all the instructions \ above. The instruction should begin with \ and end with \. The instruction should be concise, effective, and generally applicable to all problems above.

---

Figure 15: An example of the meta-prompt for prompt optimization with GPT models (`gpt-3.5-turbo` or `gpt-4`) on GSM8K, where the generated instruction will be prepended to the beginning of the scorer LLM output (*A_begin* in Section 4). The blue text contains solution-score pairs; the purple text describes the optimization task and output format; the orange text are meta-instructions.

## F  PROMPT OPTIMIZATION ON GSM8K: MORE RESULTS

We first present the results of generating Q_begin instructions with the `text-bison` scorer and the `PaLM 2-L-IT` optimizer, starting from an empty instruction with a 57.1 training accuracy. The optimization curve in Figure 16(a) shows a similar upward trend as in Figure 1(a) in the main paper. Table 5 summarizes top instructions, and the characteristics of the instruction generated by each model are similar to those in Table 1 in the main paper.

Then, Figure 16(b) shows that the pre-trained `PaLM 2-L` can also serve as the optimizer LLM and improve its own prediction performance. Different from other optimizer LLMs that are instruction-tuned, the pre-trained `PaLM 2-L` performs better when the prompt is formatted in a few-shot manner. Therefore, we include two initial instructions to start the optimization: the empty instruction (with a training accuracy 32.2) and "The answer is" (with a training accuracy 33.3). See Figure 14 in Appendix E for the meta-prompt format. The generated instructions follow the same style as "The answer is": most instructions are also phrases suitable as the prefix of a sentence, like "Here you go:" (generated at Step 11 with training accuracy 61.3) and "Let's do it:" (generated at Step 13 with training accuracy 75.1).

Table 5: Test accuracies on GSM8K with the `text-bison` scorer.

| Optimizer / Source | Instruction position | Top instruction | Acc |
|---|---|---|---|
| *Baselines* | | | |
| (Kojima et al., 2022) | Q_begin | Let's think step by step. | 64.4 |
| (Zhou et al., 2022b) | Q_begin | Let's work this out in a step by step way to be sure we have the right answer. | 65.6 |
| | Q_begin | Let's solve the problem. | 59.1 |
| | Q_begin | (empty string) | 56.8 |
| *Ours* | | | |
| `PaLM 2-L-IT` | Q_begin | Let's work together to solve math word problems! First, we will read and discuss the problem together to make sure we understand it. Then, we will work together to find the solution. I will give you hints and help you work through the problem if you get stuck. | 64.4 |
| `text-bison` | Q_end | Let's work through this problem step-by-step: | **68.5** |
| `gpt-3.5-turbo` | Q_end | Analyze the given information, break down the problem into manageable steps, apply suitable mathematical operations, and provide a clear, accurate, and concise solution, ensuring precise rounding if necessary. Consider all variables and carefully consider the problem's context for an efficient solution. | 66.5 |
| `gpt-4` | Q_begin | Start by dissecting the problem to highlight important numbers and their relations. Decide on the necessary mathematical operations like addition, subtraction, multiplication, or division, required for resolution. Implement these operations, keeping in mind any units or conditions. Round off by ensuring your solution fits the context of the problem to ensure accuracy. | 62.7 |

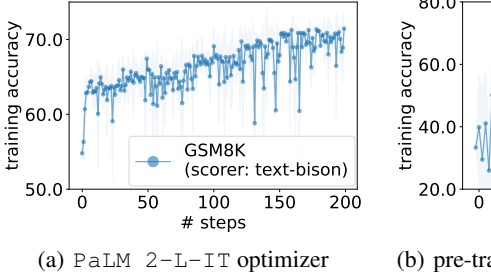
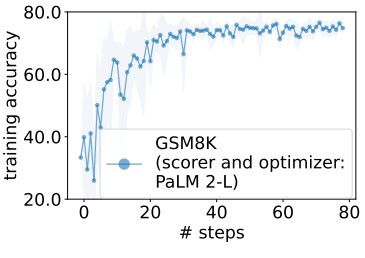

(a) `PaLM 2-L-IT` optimizer      (b) pre-trained `PaLM 2-L` optimizer

Figure 16: Prompt optimization on GSM8K with (a) the `text-bison` scorer and the `PaLM 2-L-IT` optimizer, and (b) pre-trained `PaLM 2-L` as both scorer and optimizer.

## G    PROMPT OPTIMIZATION: INSTRUCTIONS FOUND IN INTERMEDIATE STEPS

### G.1    GSM8K, A_BEGIN INSTRUCTIONS, PRE-TRAINED PaLM 2-L AS SCORER, PaLM 2-L-IT AS OPTIMIZER, STARTING FROM "LET'S SOLVE THE PROBLEM"

A few leaps in the training accuracy include:

- "Let's think carefully about the problem and solve it together." at Step 2 with the training accuracy 63.2;
- "Let's break it down!" at Step 4 with training accuracy 71.3;
- "Let's calculate our way to the solution!" at Step 5 with training accuracy 73.9;
- "Let's do the math!" at Step 6 with training accuracy 78.2.

### G.2    GSM8K, Q_BEGIN INSTRUCTIONS, TEXT-BISON AS SCORER, PaLM 2-L-IT AS OPTIMIZER, STARTING FROM EMPTY STRING

A few leaps in the training accuracy include:

- "Solve the following problems using the given information." at Step 2 with training accuracy 59.8;
- "Solve the following problems by applying the given information and using the appropriate mathematical operations." at Step 3 with training accuracy 64.0;
- "Let's read the problem carefully and identify the given information. Then, we can create an equation and solve for the unknown variable." at Step 4 with training accuracy 67.0;
- "I'm always down for solving a math word problem together. Just give me a moment to read and understand the problem. Then, I'll create an equation that models the problem, which I'll solve for the unknown variable. I also may or may not use some helpful diagrams or visuals to understand the problem. Lastly, be sure to allow me some time to carefully check my work before submitting any responses!" at Step 29 with training accuracy 70.1.

## H  SEMANTICALLY SIMILAR INSTRUCTIONS MAY ACHIEVE DRASTICALLY DIFFERENT ACCURACIES

One challenge of prompt optimization is the sensitivity of model performance to subtle changes in the instruction. For example, with the `PaLM 2-L` scorer on the GSM8K test set, "Let's think step by step." achieves accuracy 71.8, "Let's solve the problem together." has accuracy 60.5, while the accuracy of "Let's work together to solve this problem step by step." is only 49.4, although it is the semantic combination of the two upper instructions. This behavior increases both the variance across single-step instructions and the oscillation during optimization, and motivates us to generate multiple instructions at each step to improve the optimization stability.

# I    PROMPT OPTIMIZATION CURVES ON 23 BBH TASKS

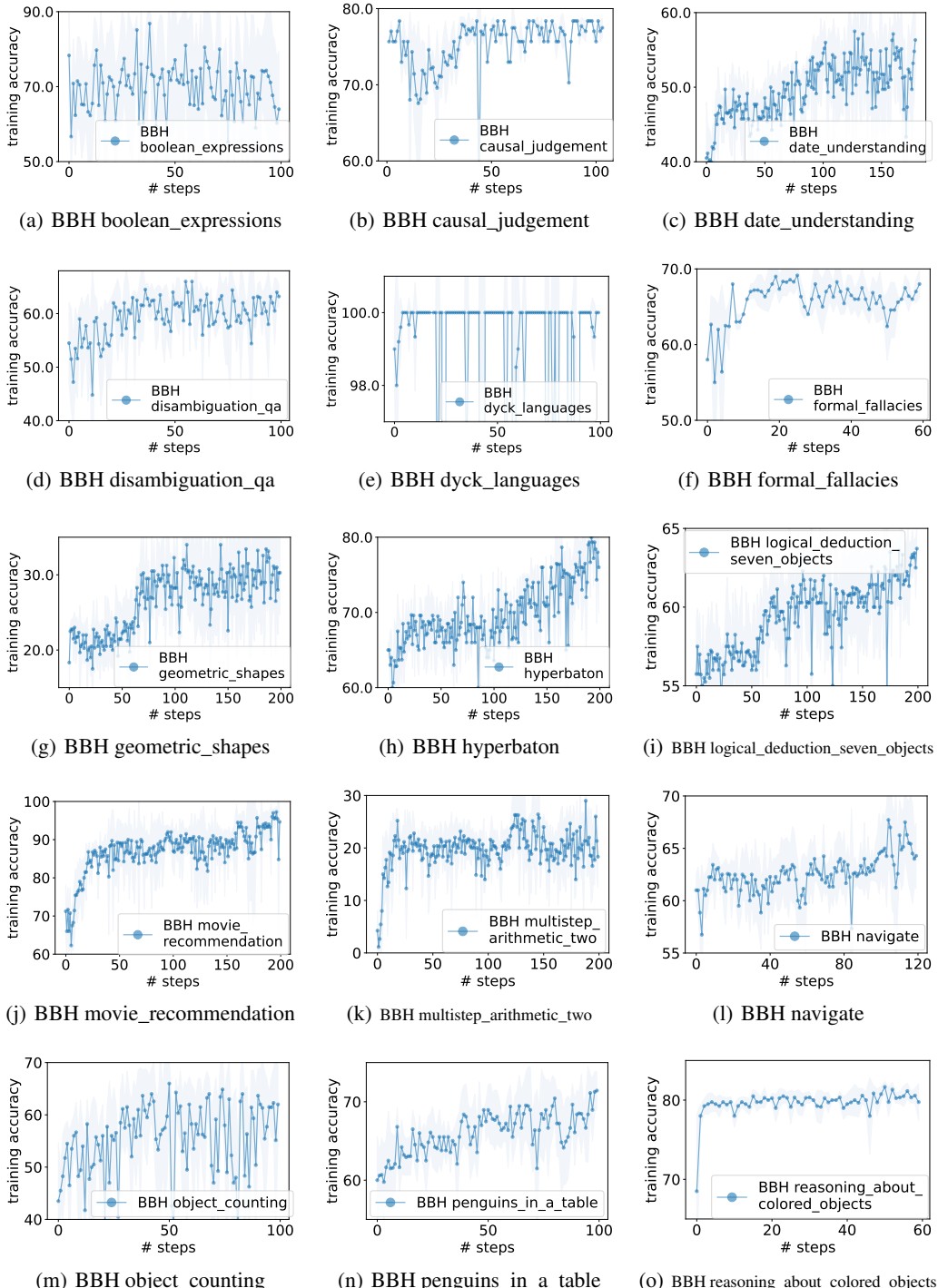

Figure 17: (Part 1/2) Prompt optimization on 23 BBH tasks with the `text-bison` scorer and the `PaLM 2-L-IT` optimizer. The optimizations start from the empty string. Most curves have upward trends.

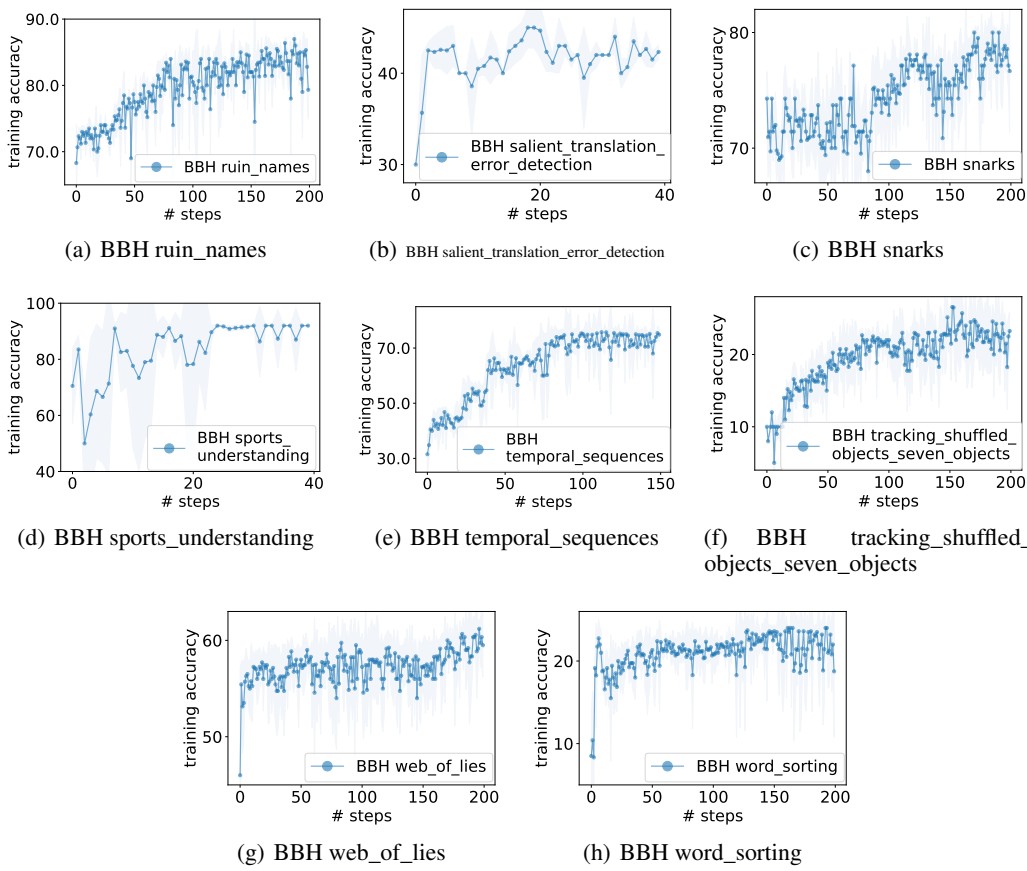

(a) BBH ruin_names  (b) BBH salient_translation_error_detection  (c) BBH snarks

(d) BBH sports_understanding  (e) BBH temporal_sequences  (f) BBH tracking_shuffled_objects_seven_objects

(g) BBH web_of_lies  (h) BBH word_sorting

Figure 18: (Part 2/2) Prompt optimization on 23 BBH tasks with the `text-bison` scorer and the `PaLM 2-L-IT` optimizer. The optimizations start from the empty string. Most curves have upward trends.

## J  PROMPT OPTIMIZATION ON BBH TASKS – TABULATED ACCURACIES AND FOUND INSTRUCTIONS

### J.1  PALM 2-L-IT AS OPTIMIZER, OPTIMIZATION STARTING FROM THE EMPTY STRING

Table 7 and 8 show the instructions found by prompt optimization. A comparison of their accuracy values with baselines "Let's think step by step." (Kojima et al., 2022), "Let's work this out in a step by step way to be sure we have the right answer." (Zhou et al., 2022b), and the empty string is in Table 6. A bar chart visualization of the performance difference with the `PaLM 2-L` scorer is at Figure 19(a) and 19(b) (which are the same as Figure 4(a) and 4(b) in the main paper but have BBH task names); a bar chart visualization with the `text-bison` scorer is at Figure 19(c) and 19(d).

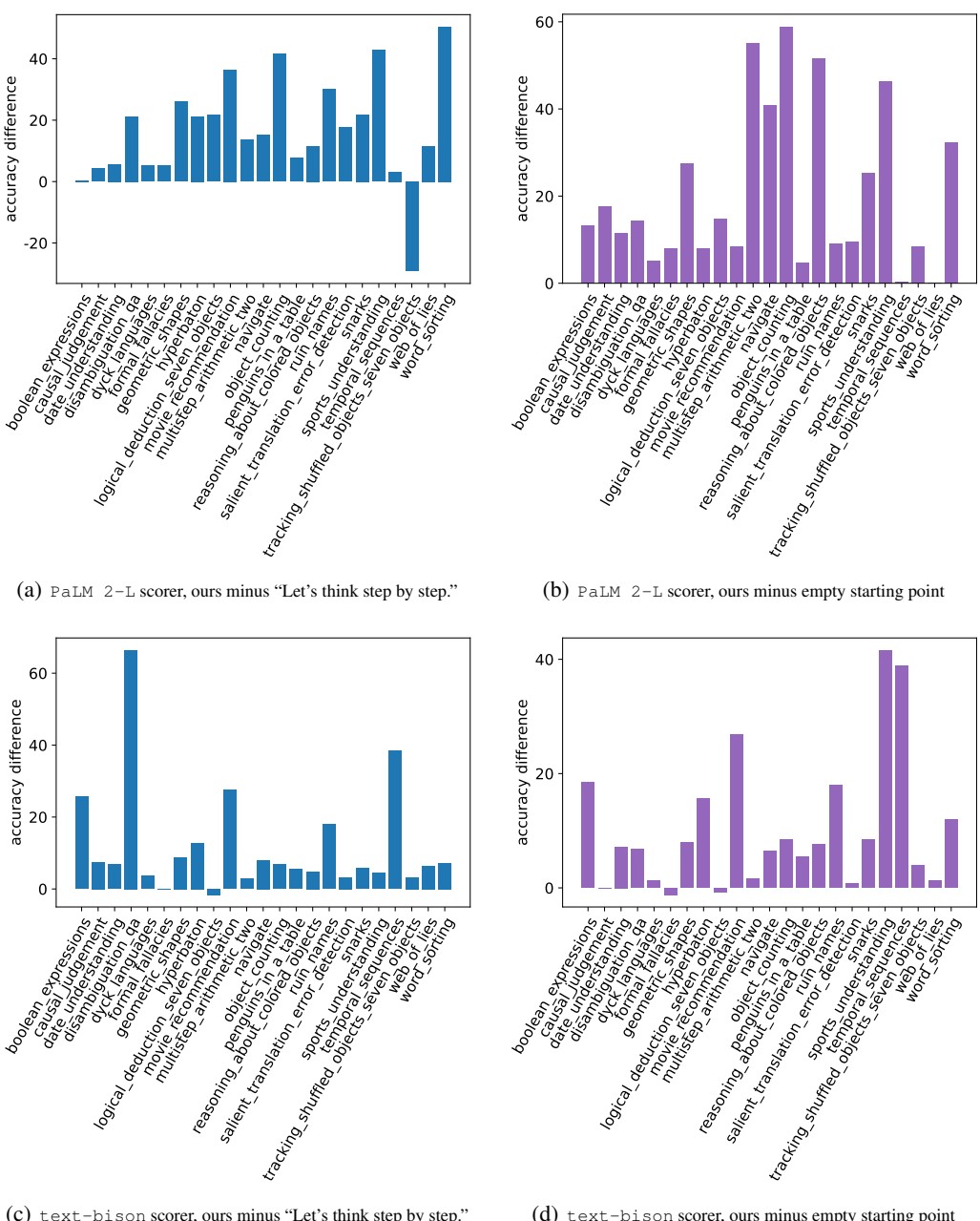

(a) `PaLM 2-L` scorer, ours minus "Let's think step by step."

(b) `PaLM 2-L` scorer, ours minus empty starting point

(c) `text-bison` scorer, ours minus "Let's think step by step."

(d) `text-bison` scorer, ours minus empty starting point

Figure 19: On 23 BBH tasks, the accuracy differences among instructions found by prompt optimization (with the `PaLM 2-L-IT` optimizer), "Let's think step by step.", and the empty string (optimization starting point).

Table 6: Accuracies on BBH tasks: our found instructions with the `PaLM 2-L-IT` optimizer vs baseline. The optimization starts from the empty string. Because of the 20-80 train-test split, we show accuracies with the format "training / test / overall (training + test)". The `PaLM 2-L` scores are from A_begin instructions; the `text-bison` scores are from Q_begin instructions. Bold numbers indicate the best for the corresponding task.

| Task | Scorer | Our Acc | "Let's think step by step." Acc | "Let's work this out in a step by step way to be sure we have the right answer." Acc | empty string "" Acc |
|---|---|---|---|---|---|
| | | training / test / overall | training / test / overall | training / test / overall | training / test / overall |
| boolean_expressions | PaLM 2-L | **90.0 / 83.5 / 84.8** | 90.0 / 83.0 / 84.4 | 82.0 / 74.0 / 75.6 | 74.0 / 71.0 / 71.6 |
| causal_judgement | PaLM 2-L | **84.8 / 58.0 / 63.1** | 73.0 / 55.3 / 58.8 | 59.5 / 57.3 / 57.8 | 29.7 / 49.3 / 45.5 |
| date_understanding | PaLM 2-L | **86.0 / 84.5 / 84.8** | 76.0 / 80.0 / 79.2 | 74.0 / 77.0 / 76.4 | 70.0 / 74.0 / 73.2 |
| disambiguation_qa | PaLM 2-L | **80.0 / 69.0 / 71.2** | 40.0 / 52.5 / 50.0 | 48.0 / 47.0 / 47.2 | 54.0 / 57.5 / 56.8 |
| dyck_languages | PaLM 2-L | **100.0 / 100.0 / 100.0** | 96.0 / 94.5 / 94.8 | 100.0 / 93.5 / 94.8 | 94.0 / 95.0 / 94.8 |
| formal_fallacies | PaLM 2-L | **84.0 / 64.0 / 68.4** | 78.0 / 59.5 / 63.2 | 68.0 / 63.0 / 64.0 | 66.0 / 59.0 / 60.4 |
| geometric_shapes | PaLM 2-L | **76.0 / 57.0 / 60.8** | 42.0 / 33.0 / 34.8 | 42.0 / 32.0 / 34.0 | 34.0 / 33.0 / 33.2 |
| hyperbaton | PaLM 2-L | **100.0 / 96.0 / 96.8** | 78.0 / 75.0 / 75.6 | 74.0 / 72.5 / 72.8 | 88.0 / 89.0 / 88.8 |
| logical_deduction_seven_objects | PaLM 2-L | **74.0 / 57.0 / 60.4** | 46.0 / 37.0 / 38.8 | 34.0 / 30.5 / 31.2 | 46.0 / 45.5 / 45.6 |
| movie_recommendation | PaLM 2-L | **92.0 / 90.5 / 90.8** | 62.0 / 52.5 / 54.4 | 52.0 / 48.0 / 48.8 | 80.0 / 83.0 / 82.4 |
| multistep_arithmetic_two | PaLM 2-L | **72.0 / 55.5 / 58.8** | 42.0 / 46.0 / 45.2 | 60.0 / 50.5 / 52.4 | 4.0 / 3.5 / 3.6 |
| navigate | PaLM 2-L | **92.0 / 75.0 / 78.4** | 68.0 / 62.0 / 63.2 | 70.0 / 64.0 / 65.2 | 38.0 / 37.5 / 37.6 |
| object_counting | PaLM 2-L | **84.0 / 86.5 / 86.0** | 36.0 / 46.5 / 44.4 | 60.0 / 62.0 / 61.6 | 28.0 / 27.0 / 27.2 |
| penguins_in_a_table | PaLM 2-L | **86.2 / 71.8 / 74.7** | 79.3 / 64.1 / 67.1 | 62.1 / 58.1 / 58.9 | 72.4 / 69.2 / 69.9 |
| reasoning_about_colored_objects | PaLM 2-L | **98.0 / 85.5 / 88.0** | 82.0 / 79.5 / 80.0 | 82.0 / 75.0 / 76.4 | 42.0 / 35.0 / 36.4 |
| ruin_names | PaLM 2-L | **88.0 / 88.0 / 88.0** | 70.0 / 55.0 / 58.0 | 80.0 / 75.5 / 76.4 | 88.0 / 76.5 / 78.8 |
| salient_translation_error_detection | PaLM 2-L | **62.0 / 67.0 / 66.0** | 42.0 / 50.0 / 48.4 | 58.0 / 46.0 / 48.4 | 56.0 / 56.5 / 56.4 |
| snarks | PaLM 2-L | **85.7 / 83.2 / 83.7** | 60.0 / 62.2 / 61.8 | 54.3 / 53.1 / 53.4 | 51.4 / 60.1 / 58.4 |
| sports_understanding | PaLM 2-L | **98.0 / 88.0 / 90.0** | 50.0 / 46.5 / 47.2 | 60.0 / 52.5 / 54.0 | 52.0 / 41.5 / 43.6 |
| temporal_sequences | PaLM 2-L | **100.0 / 100.0 / 100.0** | 100.0 / 96.0 / 96.8 | 90.0 / 87.0 / 87.6 | 100.0 / 99.5 / 99.6 |
| tracking_shuffled_objects_seven_objects | PaLM 2-L | 32.0 / 16.5 / 19.6 | **58.0 / 61.5 / 60.8** | 54.0 / 55.5 / 55.2 | 14.0 / 23.5 / 21.6 |
| web_of_lies | PaLM 2-L | 62.0 / 52.0 / 54.0 | 46.0 / 41.5 / 42.4 | 24.0 / 31.0 / 29.6 | **54.0 / 54.0 / 54.0** |
| word_sorting | PaLM 2-L | **54.0 / 54.5 / 54.4** | 2.0 / 4.5 / 4.0 | 12.0 / 9.5 / 10.0 | 20.0 / 22.5 / 22.0 |
| boolean_expressions | text-bison | **98.0 / 87.0 / 89.2** | 72.0 / 61.5 / 63.6 | 88.0 / 78.0 / 80.0 | 80.0 / 68.5 / 70.8 |
| causal_judgement | text-bison | 78.4 / 58.0 / 62.0 | 70.3 / 50.7 / 54.5 | 73.0 / 55.3 / 58.8 | **78.4 / 58.0 / 62.0** |
| date_understanding | text-bison | **60.0 / 50.0 / 52.0** | 44.0 / 45.5 / 45.2 | 48.0 / 45.0 / 45.6 | 44.0 / 45.0 / 44.8 |
| disambiguation_qa | text-bison | **68.0 / 73.0 / 72.0** | 4.0 / 6.0 / 5.6 | 4.0 / 15.5 / 13.2 | 52.0 / 68.5 / 65.2 |
| dyck_languages | text-bison | **100.0 / 100.0 / 100.0** | 100.0 / 95.5 / 96.4 | 100.0 / 94.5 / 95.6 | 100.0 / 98.5 / 98.8 |
| formal_fallacies | text-bison | 70.0 / 53.0 / 56.4 | 64.0 / 54.5 / 56.4 | **84.0 / 82.5 / 82.8** | 70.0 / 54.5 / 57.6 |
| geometric_shapes | text-bison | **40.0 / 19.5 / 23.6** | 22.0 / 13.0 / 14.8 | 18.0 / 12.0 / 13.2 | 20.0 / 14.5 / 15.6 |
| hyperbaton | text-bison | **80.0 / 79.5 / 79.6** | 64.0 / 67.5 / 66.8 | 64.0 / 69.0 / 68.0 | 64.0 / 64.0 / 64.0 |
| logical_deduction_seven_objects | text-bison | 66.0 / 53.5 / 56.0 | **56.0 / 58.0 / 57.6** | 56.0 / 56.0 / 56.0 | 58.0 / 56.5 / 56.8 |
| movie_recommendation | text-bison | **98.0 / 90.0 / 91.6** | 68.0 / 63.0 / 64.0 | 66.0 / 62.0 / 62.8 | 68.0 / 64.0 / 64.8 |
| multistep_arithmetic_two | text-bison | **32.0 / 16.5 / 19.6** | 12.0 / 18.0 / 16.8 | 18.0 / 17.5 / 17.6 | 16.0 / 18.5 / 18.0 |
| navigate | text-bison | **72.0 / 61.0 / 63.2** | 56.0 / 55.0 / 55.2 | 60.0 / 56.5 / 57.2 | 56.0 / 57.0 / 56.8 |
| object_counting | text-bison | **72.0 / 62.0 / 64.0** | 58.0 / 57.0 / 57.2 | 62.0 / 55.5 / 56.8 | 50.0 / 57.0 / 55.6 |
| penguins_in_a_table | text-bison | **72.4 / 56.4 / 59.6** | 58.6 / 53.0 / 54.1 | 55.2 / 55.6 / 55.5 | 58.6 / 53.0 / 54.1 |
| reasoning_about_colored_objects | text-bison | **82.0 / 77.0 / 78.0** | 76.0 / 72.5 / 73.2 | 78.0 / 73.0 / 74.0 | 74.0 / 69.5 / 70.4 |
| ruin_names | text-bison | **88.0 / 82.5 / 83.6** | 66.0 / 65.5 / 65.6 | 66.0 / 62.5 / 63.2 | 64.0 / 66.0 / 65.6 |
| salient_translation _error_detection | text-bison | 46.0 / 50.5 / 49.6 | 42.0 / 47.5 / 46.4 | 42.0 / 49.5 / 48.0 | 44.0 / 50.0 / 48.8 |
| snarks | text-bison | **80.0 / 81.8 / 81.5** | 68.6 / 77.6 / 75.8 | 71.4 / 76.2 / 75.3 | 77.1 / 84.6 / 73.1 |
| sports_understanding | text-bison | **94.0 / 82.5 / 84.8** | 86.0 / 79.0 / 80.4 | 90.0 / 81.0 / 82.8 | 38.0 / 44.5 / 43.2 |
| temporal_sequences | text-bison | 78.0 / 81.0 / 80.4 | 36.0 / 43.5 / 42.0 | 32.0 / 45.0 / 42.4 | 36.0 / 43.0 / 41.6 |
| tracking_shuffled_objects_seven_objects | text-bison | **32.0 / 15.5 / 18.8** | 10.0 / 17.0 / 15.6 | 10.0 / 18.0 / 16.4 | 12.0 / 15.5 / 14.8 |
| web_of_lies | text-bison | **62.0 / 50.0 / 52.4** | 48.0 / 45.5 / 46.0 | 48.0 / 44.0 / 44.8 | 52.0 / 51.5 / 51.2 |
| word_sorting | text-bison | **24.0 / 17.5 / 18.8** | 10.0 / 12.0 / 11.6 | 4.0 / 8.0 / 7.2 | 4.0 / 7.5 / 6.8 |

Table 7: BBH task-wise instructions found by prompt optimization with the `PaLM 2-L` scorer and the `PaLM 2-L-IT` optimizer. The optimization starts from the empty string.

| Task | Our Instruction |
|---|---|
| boolean_expressions | A Boolean expression is a well-formed expression consisting of variables, values, and logical operators. The expression must evaluate to a single True or False value. The order of precedence of the logical operators is as follows: NOT, AND, OR, XOR, IMP. Parentheses can be used to group subexpressions and to control the order of evaluation. |
| causal_judgement | When considering questions about causation, a typical person would consider the following factors: whether the action or event was a necessary condition for the outcome to occur, a sufficient condition, a proximate cause, or a foreseeable cause. |
| date_understanding | To find the date X time ago from today, first find today's date. Then subtract X time from today's date. If the current date is the last day of a month, then the date a month ago is the last day of the previous month. If the current date is not the last day of a month, then the date a month ago is the same day of the previous month. For example, if today is March 31, 2023, then the date a month ago is February 28, 2023. If today is April 1, 2023, then the date a month ago is March 1, 2023. |
| disambiguation_qa | Identifying Antecedents of Pronouns: A Comprehensive Guide |
| dyck_languages | First, look for the opening parentheses. Then, count the number of opening parentheses. Finally, close the parentheses in the reverse order that they were opened. |
| formal_fallacies | A deductive argument is one where the conclusion follows necessarily from the premises. If the premises are true, then the conclusion must also be true. An invalid argument is one where it is possible for the premises to be true and the conclusion to be false. |
| geometric_shapes | A closed polygonal chain is a series of connected line segments. The line segments can be straight or curved. The first and last line segments are connected. The line segments do not intersect each other except at their endpoints. A closed polygon can be described by an SVG path element, which starts at a given point, goes to one or more additional points, and then ends at the starting point. The path element can consist of straight line segments, curved segments, or a mixture of both. |
| hyperbaton | The correct adjective order in English is opinion, size, shape, age, color, origin, material, and purpose. If you have more than one adjective of the same type, they are usually placed in order of importance. For example, you would say "a large, old, Pakistani ship" rather than "an old, large, Pakistani ship." There are a few exceptions to these rules, but they are generally followed in most cases. |
| logical_deduction _seven_objects | The following questions will test your ability to use deductive reasoning. You will be given a set of statements about a group of objects. You will then be asked to answer questions about the objects based on the statements. The statements in the questions are logically consistent, so you can use them to deduce the order of the objects. For each question, you must choose the option that is logically consistent with the information in the questions. |
| movie_recommendation | Based on your input, I have analyzed the given movies in terms of genre, plot, tone, audience rating, year of release, director, cast, and reviews. I have also taken into account the given options. The movie that is most similar to the given movies in terms of all these factors is: |
| multistep_arithmetic _two | The order of operations in mathematics is PEMDAS, which stands for Parentheses, Exponents, Multiplication, Division, Addition, and Subtraction. When there are multiple operations of the same precedence, they must be performed from left to right. Note that multiplication and division have the same precedence, as do addition and subtraction. |
| navigate | You will return to the starting point if and only if (1) the total number of steps you take forward is equal to the total number of steps you take back, and (2) the total number of turns you make is a multiple of 180 degrees. |
| object_counting | Here is a list of the objects you mentioned and their corresponding counts: |
| penguins_in_a_table | Here is my new text: |
| reasoning_about _colored_objects | Starting from the leftmost object in the row, I observe the following objects arranged in this order: |
| ruin_names | Which is the funniest pun on the artist or movie name? |
| salient_translation _error_detection | Instructions: Read the German sentence and its English translation carefully, then identify the type of error in the translation and select the correct option. There are six possible types of errors: Named Entities, Numerical Values, Modifiers or Adjectives, Negation or Antonyms, Facts, and Dropped Content. |
| snarks | Identify the sarcastic statement by considering the following factors: incongruity, exaggeration, understatement, context, speaker's intent, and audience's reaction. I will also consider the speaker's tone of voice, facial expressions, and body language. |
| sports_understanding | I will determine if a sentence about an athlete is plausible by first checking if it is grammatically correct. If it is, I will then check if it is consistent with the athlete's sport, position, and real-world statistics. I will also check if it is consistent with the rules of the athlete's sport. If the sentence is consistent with all of these things, I will answer "yes", otherwise I will answer "no". |
| temporal_sequences | The answer is the time that is not mentioned in the given statements. |
| tracking_shuffled_objects _seven_objects | Claire has the blue ball, Gertrude has the black ball, and Dave has the green ball. They are all happy with their new balls. |
| web_of_lies | The answer to a question is yes if there are an odd number of liars before the current speaker, and no if there are an even number of liars before the current speaker. If the current speaker is a truth-teller, they will say the opposite of what the previous person said, while a liar will say the same thing as the previous person said. |
| word_sorting | Alphabetical order of given words: |

Table 8: BBH task-wise instructions found by prompt optimization with the `text-bison` scorer and the `PaLM 2-L-IT` optimizer. The optimization starts from the empty string.

| Task | Our Instruction |
|---|---|
| boolean_expressions | Not (not False) and not not False is False |
| causal_judgement | A typical person would likely answer the questions about causation as follows: |
| date_understanding | Today is February 28, 2023. It is a Tuesday. Yesterday was Monday, February 27, 2023. Tomorrow will be Wednesday, March 1, 2023. A week ago, it was February 21, 2023, and a month ago, it was January 28, 2023. A year from now, it will be February 28, 2024. The day of the week is important to note because it will help us to correctly answer the questions below. Not all years are leap years that contain February 29. |
| disambiguation_qa | A pronoun is a word that stands in for a noun. The noun that a pronoun refers to is called its antecedent. To identify the antecedent of a pronoun, look for the noun that the pronoun could be referring to. If there is only one possible noun, then that is the antecedent. If there are two or more possible nouns, then the antecedent is ambiguous. Use the context of the sentence to help you determine the correct antecedent. |
| dyck_languages | { } |
| formal_fallacies | How to Evaluate Deductive Validity of an Argument |
| geometric_shapes | What shape is this SVG code drawing, and how many sides does it have? |
| hyperbaton | In English, adjectives are typically placed before nouns in a specific order. The order is: opinion, size, shape, age, color, origin, material, purpose, noun. For example, the sentence "the big, old, red barn" would be considered grammatically correct, while the sentence "the old, big, red barn" would not. Adjectives that come before nouns are called attributive adjectives, while adjectives that come after nouns are called predicative adjectives. |
| logical_deduction _seven_objects | In this logical reasoning task, you will be given a series of paragraphs, each of which describes a set of objects arranged in a fixed order. The statements in each paragraph are logically consistent. You must read each paragraph carefully and use the information given to determine the logical relationships between the objects. You will then be asked a question about the order of the objects. Read each question carefully and choose the option that answers the question correctly. |
| movie_recommendation | What is the highest-rated movie similar to the given movies, with a similar IMDb rating and released in the same year? |
| multistep_arithmetic_two | Let's solve these equations using PEMDAS order of operations. Remember that PEMDAS stands for parentheses, exponents, multiplication and division, and addition and subtraction. |
| navigate | Starting at the origin, facing north, follow the instructions. If your displacement from the origin is zero and your direction is unchanged, then your answer is Yes. Otherwise, your answer is No. |
| object_counting | Let me help you count the items you have. Just list them one by one, separated by commas. I will then count each item and tell you how many items there are in total. |
| penguins_in_a_table | This table shows information about penguins. The columns show the penguin's name, age, height (in cm), and weight (in kg). The penguins are listed in order of their age, from youngest to oldest. |
| reasoning_about _colored_objects | First, read the input carefully. Then, identify all the objects mentioned, their colors, and their positions. Next, visualize the objects and their positions in your mind. Finally, answer the questions accurately based on the information given. Make sure to pay attention to the order of the objects. |
| ruin_names | A humorous edit of an artist or movie name can be created by replacing one or more letters to form a new word or phrase that sounds similar but has a different meaning. The new word or phrase should be relevant to the original word, but it should also be a surprise, which makes the edit funny. For example, the artist or movie name "Rocky" can be changed to "Ricky," and "Schindler's List" can be changed to "Schindler's Lift." Be creative and have fun! |
| salient_translation _error_detection | The following translations from German to English contain a particular error. The error may be one of the following types: Named Entities, Numerical Values, Modifiers or Adjectives, Negation or Antonyms, Facts, or Dropped Content. Please identify the error. |
| snarks | The statement |
| sports_understanding | To determine the plausibility of a sports sentence, I will first identify the sport, athletes, teams, and events mentioned in the sentence. Then, I will use my knowledge of the rules of the sport, the context of the sentence, common sense, and my knowledge of the world to determine whether the sentence is plausible. I will also consider the time period and location, as well as any other relevant information. Finally, I will return a score of 1 for plausible sentences and 0 for implausible ones. |
| temporal_sequences | To determine the time period when a person went to a place, first identify all the time periods when the person's whereabouts are unknown. Then, rule out any time periods during which the person was seen doing something else or the place was closed. The remaining time periods are the possible times when the person could have gone to the place. |
| tracking_shuffled_objects _seven_objects | At the start of the game, Claire has a blue ball. Throughout the game, pairs of people swap balls. Claire ends up with the yellow ball. |
| web_of_lies | People in a group either tell the truth or lie. The truthfulness of a person's statement is determined by the statement of the previous person. If the previous person told the truth, then the current person who says the opposite is lying. If the previous person lied, then the current person who says the opposite is telling the truth. This rule applies to all subsequent statements. |
| word_sorting | Sort the following words alphabetically, ignoring case and punctuation. Print the sorted list. |

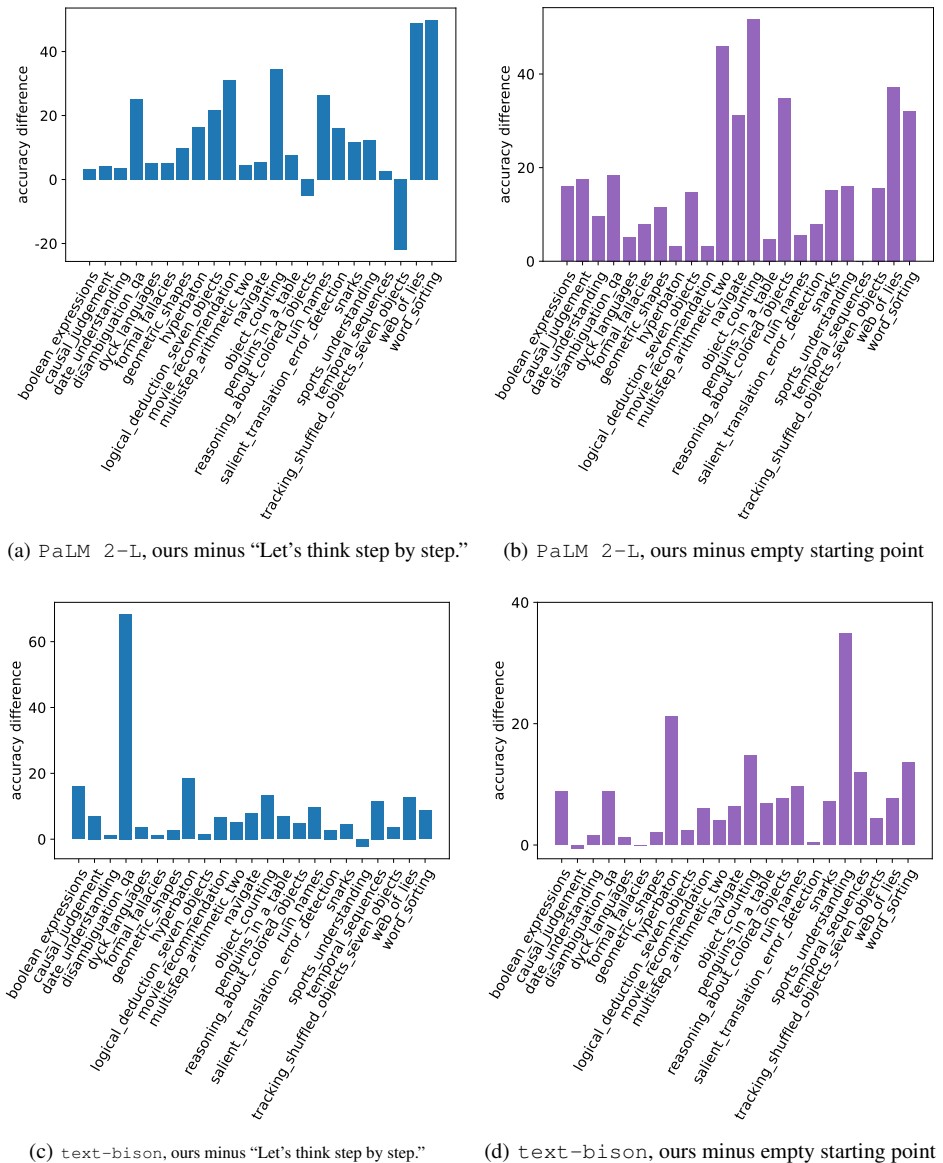

(a) `PaLM 2-L`, ours minus "Let's think step by step."    (b) `PaLM 2-L`, ours minus empty starting point

(c) `text-bison`, ours minus "Let's think step by step."    (d) `text-bison`, ours minus empty starting point

Figure 20: On 23 BBH tasks, the accuracy differences among instructions found by prompt optimization (with the `gpt-3.5-turbo` optimizer), "Let's think step by step.", and the empty string (optimization starting point).

## J.2 `GPT-3.5-TURBO` AS OPTIMIZER, OPTIMIZATION STARTING FROM THE EMPTY STRING

Table 10, 11 and 12 show the instructions found by prompt optimization. Their accuracies are listed in Table 9. Figure 20 visualizes the difference between their accuracies and those of the baselines "Let's think step by step." and the empty string. The optimizations find instructions better than the empty starting point, and most of the found instructions are better than "Let's think step by step".

One caveat in the A_begin instructions (Table 10) is that some instructions found here are imperative or interrogative sentences that are more suitable to be put into "Q:" rather than "A:", like "Solve the sequence by properly closing the parentheses." for dyck_languages and "Which movie option from the given choices ...?" for movie_recommendation. Such styles appear more often here than the with `PaLM 2-L-IT` optimizer (Table 7), indicating `PaLM 2-L-IT` understands the desirable style

Table 9: Accuracies on BBH tasks with the `gpt-3.5-turbo` optimizer that starts from the empty string. The `PaLM 2-L` scores are from A_begin (left) instructions; the `text-bison` scores include Q_begin (left) and Q_end (right) instructions.

| Task | Scorer | Our Acc (`begin`) training / test / overall | Our Acc (`end`) training / test / overall |
|---|---|---|---|
| boolean_expressions | PaLM 2-L | 92.0 / 86.5 / 87.6 | N/A |
| causal_judgement | PaLM 2-L | 81.1 / 58.7 / 63.1 | N/A |
| date_understanding | PaLM 2-L | 86.0 / 82.0 / 82.8 | N/A |
| disambiguation_qa | PaLM 2-L | 80.0 / 74.0 / 75.2 | N/A |
| dyck_languages | PaLM 2-L | 100.0 / 100.0 / 100.0 | N/A |
| formal_fallacies | PaLM 2-L | 88.0 / 63.5 / 68.4 | N/A |
| geometric_shapes | PaLM 2-L | 60.0 / 41.0 / 44.8 | N/A |
| hyperbaton | PaLM 2-L | 88.0 / 93.0 / 92.0 | N/A |
| logical_deduction_seven_objects | PaLM 2-L | 76.0 / 56.5 / 60.4 | N/A |
| movie_recommendation | PaLM 2-L | 84.0 / 86.0 / 85.6 | N/A |
| multistep_arithmetic_two | PaLM 2-L | 52.0 / 49.0 / 49.6 | N/A |
| navigate | PaLM 2-L | 76.0 / 67.0 / 68.8 | N/A |
| object_counting | PaLM 2-L | 78.0 / 79.0 / 78.8 | N/A |
| penguins_in_a_table | PaLM 2-L | 82.8 / 72.6 / 74.7 | N/A |
| reasoning_about _colored_objects | PaLM 2-L | 86.0 / 67.5 / 71.2 | N/A |
| ruin_names | PaLM 2-L | 90.0 / 83.0 / 84.4 | N/A |
| salient_translation_error_detection | PaLM 2-L | 62.0 / 65.0 / 64.4 | N/A |
| snarks | PaLM 2-L | 85.7 / 70.6 / 73.6 | N/A |
| sports_understanding | PaLM 2-L | 68.0 / 57.5 / 59.6 | N/A |
| temporal_sequences | PaLM 2-L | 100.0 / 99.5 / 99.6 | N/A |
| tracking_shuffled_objects_seven_objects | PaLM 2-L | 44.0 / 34.5 / 36.4 | N/A |
| web_of_lies | PaLM 2-L | 92.0 / 91.0 / 91.2 | N/A |
| word_sorting | PaLM 2-L | 62.0 / 52.0 / 54.0 | N/A |
| boolean_expressions | text-bison | 84.0 / 78.5 / 79.6 | 80.0 / 78.0 / 78.4 |
| causal_judgement | text-bison | 78.4 / 57.3 / 61.5 | 83.8 / 53.3 / 59.4 |
| date_understanding | text-bison | 52.0 / 45.0 / 46.4 | 64.0 / 52.4 / 54.8 |
| disambiguation_qa | text-bison | 68.0 / 75.5 / 74.0 | 64.0 / 71.5 / 70.0 |
| dyck_languages | text-bison | 100.0 / 99.5 / 99.6 | 100.0 / 100.0 / 100.0 |
| formal_fallacies | text-bison | 70.0 / 54.5 / 57.6 | 74.0 / 53.5 / 57.6 |
| geometric_shapes | text-bison | 28.0 / 15.0 / 17.6 | 48.0 / 28.0 / 32.0 |
| hyperbaton | text-bison | 86.0 / 85.0 / 85.2 | 80.0 / 76.5 / 77.2 |
| logical_deduction_seven_objects | text-bison | 66.0 / 57.5 / 59.2 | 62.0 / 55.0 / 56.4 |
| movie_recommendation | text-bison | 76.0 / 69.5 / 70.8 | 82.0 / 70.5 / 72.8 |
| multistep_arithmetic_two | text-bison | 28.0 / 20.5 / 22.0 | 28.0 / 22.5 / 23.6 |
| navigate | text-bison | 72.0 / 61.0 / 63.2 | 68.0 / 59.5 / 61.2 |
| object_counting | text-bison | 68.0 / 71.0 / 70.4 | 72.0 / 69.0 / 69.6 |
| penguins_in_a_table | text-bison | 65.5 / 59.8 / 61.0 | 79.3 / 53.0 / 58.2 |
| reasoning_about_colored_objects | text-bison | 84.0 / 76.5 / 78.0 | 86.0 / 74.0 / 76.4 |
| ruin_names | text-bison | 80.0 / 74.0 / 75.2 | 74.0 / 75.0 / 74.8 |
| salient_translation_error_detection | text-bison | 44.0 / 50.5 / 49.2 | 48.0 / 51.0 / 50.4 |
| snarks | text-bison | 82.9 / 79.7 / 80.3 | 88.6 / 84.6 / 85.4 |
| sports_understanding | text-bison | 84.0 / 76.5 / 78.0 | 90.0 / 80.0 / 82.0 |
| temporal_sequences | text-bison | 50.0 / 54.5 / 53.6 | 64.0 / 61.5 / 62.0 |
| tracking_shuffled_objects_seven_objects | text-bison | 22.0 / 18.5 / 19.2 | 30.0 / 21.5 / 23.2 |
| web_of_lies | text-bison | 64.0 / 57.5 / 58.8 | 68.0 / 55.0 / 57.6 |
| word_sorting | text-bison | 26.0 / 19.0 / 20.4 | 32.0 / 25.5 / 26.8 |

better. In Section J.3, we show the A_begin optimization results with the non-empty starting point "Let's solve the problem.". Most results there are declarative sentences – more suitable for A_begin.

Table 10: BBH task-wise instructions found by prompt optimization with the `PaLM 2-L` scorer and the `gpt-3.5-turbo` optimizer. The optimizations start from the empty string.

| Task | Our Instruction |
|---|---|
| boolean_expressions | An accurate evaluation of logical expressions involves correctly applying Boolean operators, considering the order of operations, and analyzing the truth values of the operands in accordance with Boolean logic principles. |
| causal_judgement | Understanding causality is critical for accurately assessing cause and effect relationships in various scenarios, leading to well-informed judgments, precise conclusions, and definitive answers to questions about the outcomes involved. |
| date_understanding | What is the specific date mentioned or required in each given problem or question, taking into account all relevant information, available options, and the provided context? Please provide the accurate answer in the format MM/DD/YYYY. |
| disambiguation_qa | Accurately analyze and clarify the pronoun-antecedent relationship in the given sentences, identifying the appropriate referent to eliminate any potential confusion or ambiguity and ensure a precise understanding of the intended meaning. |
| dyck_languages | Solve the sequence by properly closing the parentheses. |
| formal_fallacies | In determining the deductive validity of arguments based on explicit premises, a meticulous analysis of the logical relationships and implications is essential for definitively establishing their soundness, confirming their validity or invalidity, and ensuring a reliable and robust assessment of the arguments at hand. |
| geometric_shapes | The SVG path element with the "d" attribute plays a crucial role in web development, allowing for the precise definition and rendering of various shapes on a webpage. |
| hyperbaton | Understanding the correct order of adjectives is crucial for constructing grammatically accurate and coherent sentences that effectively convey the intended meaning in diverse contexts while ensuring clarity, cohesion, and consistency throughout consistently and effortlessly. |
| logical_deduction _seven_objects | By conducting a meticulous analysis of the given information and ensuring logical consistency within each paragraph, we can accurately determine the precise order or ranking of the mentioned objects, allowing us to confidently and consistently identify the correct answer in every presented scenario with utmost precision and confidence. |
| movie_recommendation | Which movie option from the given choices closely matches the mentioned films in terms of themes, storylines, and characteristics, guaranteeing the highest possible similarity score among them all? |
| multistep_arithmetic_two | Evaluate the given mathematical expressions step by step to determine the correct solutions accurately. |
| navigate | Is it possible to determine, with absolute certainty, whether strictly adhering to the given instructions will unfailingly bring you back to the original starting point without any exceptions, errors, or deviations? |
| object_counting | Determine the total number of objects or entities mentioned in the given list, covering various categories and types, to accurately calculate the overall count. |
| penguins_in_a_table | From the given table, what information can we gather about the mentioned animals and their respective attributes, including names, ages, heights, and weights? |
| reasoning_about _colored_objects | By thoroughly examining the given information, accurately determine the answers for each question by considering the specific characteristics, colors, and positions of the mentioned objects. |
| ruin_names | Select the most amusing and clever alteration from the options provided for the given artist, movie, or title name, and accurately choose the correct answer to test your wit and creativity. |
| salient_translation _error_detection | Thoroughly examine the given translations from German to English and accurately identify any errors by carefully analyzing the text and selecting the appropriate option with meticulous attention to detail, precision, utmost accuracy, and comprehensive understanding of the language for precise evaluation and categorization. |
| snarks | Which option delivers the most devastatingly sarcastic response, brilliantly exposing the sheer absurdity and leaving absolutely no doubt whatsoever in all the given situations? |
| sports_understanding | Maintaining the accuracy, reliability, and integrity of sports event representation is essential for upholding the highest standards of credibility, trustworthiness, and overall quality in conveying information, without any compromise, misrepresentation, or distortion, thereby ensuring the factual accuracy of sports journalism. |
| temporal_sequences | Based on the provided timeline and observed activities, we can accurately determine the possible time range when each individual could have visited their intended destinations and answer questions about their visitation time. |
| tracking_shuffled_objects _seven_objects | An important point to note is that each person in the group starts with one specific book at the beginning of the semester. |
| web_of_lies | Analyzing the consistency and accuracy of statements provided by each person is crucial for determining the truthfulness of individuals in every scenario. |
| word_sorting | Please sort the given words in alphabetical order: The list of words to be sorted contains - |

Table 11: BBH task-wise Q_begin instructions found by prompt optimization with the `text-bison` scorer and the `gpt-3.5-turbo` optimizer. The optimizations start from the empty string.

| Task | Our Instruction |
|---|---|
| boolean_expressions | Group sub-expressions with parentheses to accurately evaluate logical operations: not, and, and finally or. Determine the resulting value as either True or False. |
| causal_judgement | Consider the intentions and actions of the individuals involved. |
| date_understanding | Determine the one-day difference in the given date and express it in the format MM/DD/YYYY. |
| disambiguation_qa | Determine the precise antecedent of the pronoun in the given sentence and select the correct option or state if it is ambiguous. |
| dyck_languages | Ensure that all opening brackets have a corresponding closing bracket, and that the closing brackets are in the correct order. |
| formal_fallacies | Thoroughly analyze the explicitly provided premises and determine the deductive validity of the argument based on all necessary conditions, implications, exclusions, and dependencies given. |
| geometric_shapes | Analyze the given SVG path element carefully and confidently select the correct option from the provided choices to accurately determine the corresponding shape. Pay close attention to the specific path details and confidently make the most suitable choice. |
| hyperbaton | Select the sentence that strictly adheres to the standard order of adjectives: opinion, size, age, shape, color, origin, material, and purpose. Ensure there are no deviations or alterations in the adjective order. Choose the option without any changes. |
| logical_deduction_seven_objects | Analyze the given information to accurately determine the precise order and ranking of the mentioned objects/people, considering their relationships, positions, and any provided comparisons, for a definitive and logical progression with maximum accuracy and efficiency. |
| movie_recommendation | Based on the movie list provided, carefully consider your preferences and make a well-informed decision. |
| multistep_arithmetic_two | First, simplify any expressions within parentheses following the correct order of operations to accurately evaluate the final answer with efficiency and precision. |
| navigate | Always face forward. Take 10 steps forward. Turn left. Take 5 steps forward. Take 3 steps backward. Finally, take 7 steps forward. Turn around and take 1 step forward. Repeat the previous sequence three times. Follow the given path precisely without any deviations. At the end, turn right and take 11 steps forward. If you follow these instructions, will you return to the starting point? Options: - Yes - No |
| object_counting | Determine the total count of mentioned vegetables accurately and state the final count as the answer. |
| penguins_in_a_table | Analyze the given table to accurately determine the required information based on the provided criteria and attributes of the penguins and giraffes. Utilize efficient problem-solving strategies to arrive at the correct answer. |
| reasoning_about_colored_objects | State the color of the object mentioned in the given arrangement with utmost accuracy. |
| ruin_names | Choose the option that offers the most clever and humorous alteration of the given artist or movie name. Let your creativity shine and select the answer that will undoubtedly bring a smile to your face! Make sure to think outside the box! |
| salient_translation_error_detection | Analyze the translation and accurately identify the specific error type based on the source text, providing the most appropriate corresponding option. |
| snarks | Choose the option that wickedly embodies sarcasm. |
| sports_understanding | Determine the plausibility of the given statement by evaluating factual accuracy, logical consistency, and contextual relevance, then provide a succinct and well-justified response. |
| temporal_sequences | Identify the optimal time slot for the individual to engage in the mentioned location/activity considering the given sightings and waking up time, taking into account the opening and closing times of the location and the duration of each event. |
| tracking_shuffled_objects_seven_objects | Pay attention to the given information and track the swaps/exchanges carefully to accurately determine the final possession/position/outcome for the specified individual. |
| web_of_lies | To determine the truthfulness of the last person mentioned, analyze the consistency of each statement and count the number of individuals accusing the previous person of lying. If the count of accusers is even, that person tells the truth; if it is odd, that person lies. |
| word_sorting | Alphabetically sort the given list of words, ensuring all words are included and in ascending order. |

Table 12: BBH task-wise Q_end instructions found by prompt optimization with the `text-bison` scorer and the `gpt-3.5-turbo` optimizer. The optimizations start from the empty string.

| Task | Our Instruction |
|------|-----------------|
| boolean_expressions | Accurately use order of operations and parentheses to evaluate logical expressions and determine truth values efficiently. |
| causal_judgement | Consider all relevant factors, prioritize overall well-being and ethical considerations, make well-informed decisions while foreseeing potential consequences efficiently, and consistently strive for optimal outcomes with empathy and adaptability in a thoughtful and comprehensive manner. |
| date_understanding | Subtract the specified number of days from the given date and format the outcome as MM/DD/YYYY to accurately determine the desired result in an efficient manner. |
| disambiguation_qa | Clearly identify and select the unambiguous antecedent for the pronoun or designate it as "Ambiguous" if it is unclear. |
| dyck_languages | Add the missing closing parentheses. |
| formal_fallacies | Determine the deductive validity of the argument presented based on the explicitly stated premises and reach a definitive conclusion. |
| geometric_shapes | Analyzing the given SVG path element, accurately determine its shape by closely examining its curves and coordinates, then select the correct option. |
| hyperbaton | Choose the option with the correct adjective order in each sentence, prioritizing specific attributes like size, color, and origin. Place the most specific adjective before the more general ones for precise and standardized ordering across all examples. Ensure accurate alignment of the adjectives based on their respective attributes for consistent and standardized ordering. |
| logical_deduction _seven_objects | Determine the precise order of the given objects/participants based on the provided information and establish the final ranking accurately, considering all relevant factors, while maintaining logical consistency with maximum efficiency. |
| movie_recommendation | Choose the most similar option from the choices provided that closely aligns with the given movies' themes, genres, and impact for the most accurate recommendation possible. Make your selection wisely. |
| multistep_arithmetic_two | Carefully follow the order of operations to precisely simplify the expressions within parentheses and efficiently find the accurate final answer. |
| navigate | Always face forward. Take 10 steps forward. Turn right and walk for 5 steps. Then, make a left turn and continue for 9 steps. Proceed by walking 6 steps backward. Finally, turn around and take 200 steps. Accurately track your movements, diligently adhere to the given path, and ensure to return to the starting point without any deviations or obstacles. |
| object_counting | Determine the total count of items mentioned, including all listed items, using an efficient and concise method. State the final count as your answer. |
| penguins_in_a_table | Identify the animal with the maximum measurement (weight, age, or height) in the table and state its name and species. |
| reasoning_about _colored_objects | Determine the color of each item in the given scenario and select the correct color option from the provided choices for accurate responses, ensuring utmost precision and completeness. |
| ruin_names | Choose the option that creatively and hilariously transforms the given artist or movie name. |
| salient_translation _error_detection | Carefully analyze the translations and select the most suitable option from the given choices to rectify the specific error category, ensuring complete precision, accuracy, and faithful representation of the intended meaning, while considering all relevant information in the source text. |
| snarks | Choose the option that cleverly employs sarcasm to defy all expectations and leave everyone utterly dumbfounded, questioning the very essence of their own perception. |
| sports_understanding | Evaluate the plausibility of each given statement and provide a well-supported justification based on logical reasoning, contextual understanding, and relevant evidence to arrive at a definitive and conclusive answer. |
| temporal_sequences | Identify the possible time slot for the desired activity based on the given information and sightings, then select the correct option. |
| tracking_shuffled_objects _seven_objects | Thoroughly analyze the given scenarios, systematically consider all available information, and confidently determine the final outcome with exceptional precision and optimal efficiency, while maintaining a strategic and logical approach throughout the process. |
| web_of_lies | Examine each person's statements meticulously to accurately determine the truth and confidently identify who is telling the truth, enabling you to effectively solve the given problem. |
| word_sorting | Sort the given words alphabetically using spaces as separators while maintaining their original order and including all words. |

### J.3  PaLM 2-L AS SCORER, GPT-3.5-TURBO AS OPTIMIZER, OPTIMIZATION STARTING FROM "LET'S SOLVE THE PROBLEM."

Figure 21 and Table 13 compare the accuracies of found instructions vs "Let's solve the problem.", "Let's think step by step.", and the instructions in Table 10. Table 14 details the found instructions.

The "Let's" pattern appears more often in the found instructions because of the starting points, and the instructions are more often declarative that are more suitable for A_begin, even if some are semantically far from "Let's solve the problem". In fact, "Let's" was adopted by Zhou et al. (2022b) as a fixed pattern in generated prompts, possibly because of the same reason.

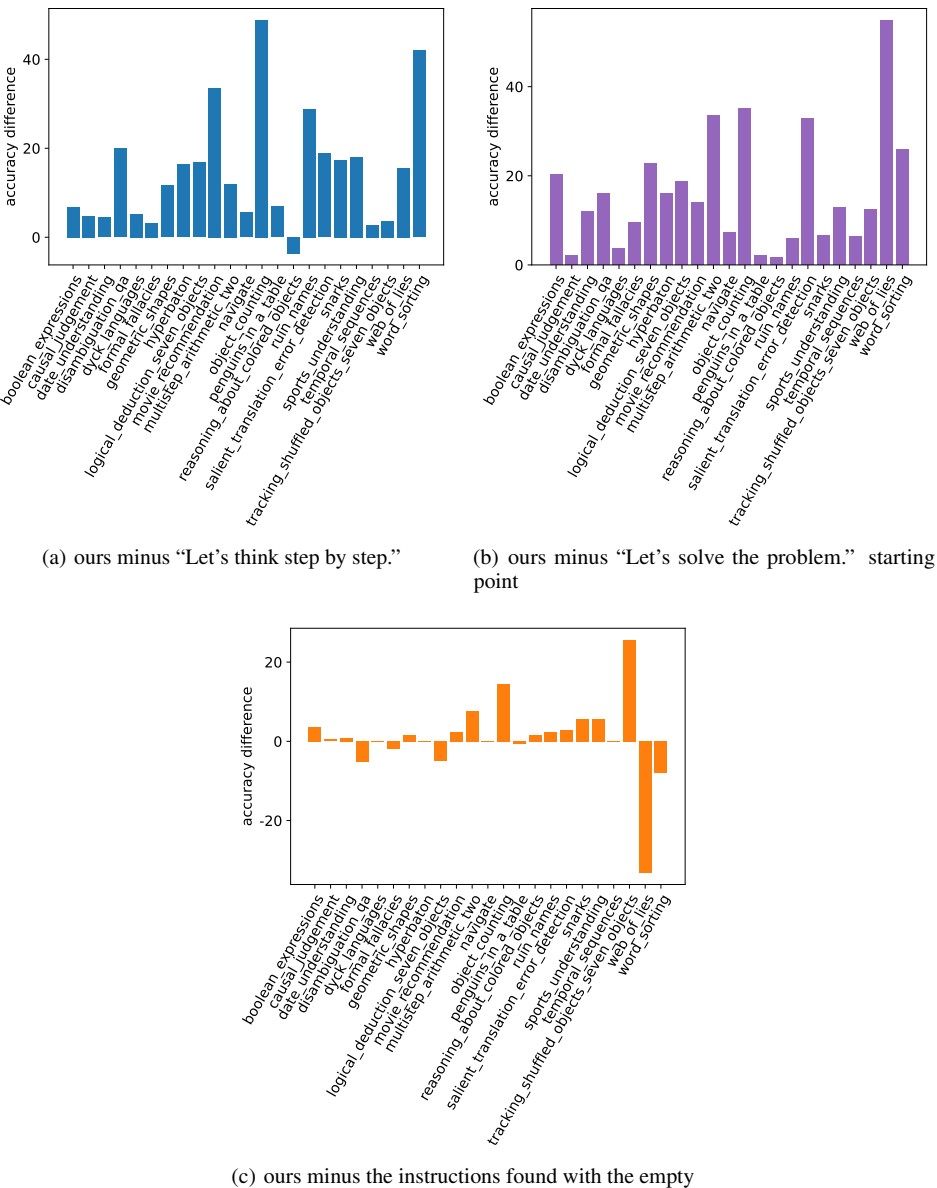

(a) ours minus "Let's think step by step."

(b) ours minus "Let's solve the problem." starting point

(c) ours minus the instructions found with the empty starting point

Figure 21: On 23 BBH tasks, the accuracy differences among instructions found by prompt optimization (with the text-bison scorer and the gpt-3.5-turbo optimizer), "Let's think step by step.", and "Let's solve the problem." (optimization starting point). The found instructions mostly outperform the "Let's think step by step." baseline, the "Let's solve the problem." starting point, and the instructions in Table 10 found by prompt optimization from the empty string.

Table 13: Accuracies on BBH tasks with the `PaLM 2-L` scorer and the `gpt-3.5-turbo` optimizer that starts from "Let's solve the problem". The scores are from A_begin instructions.

| Task | Scorer | Our Acc | "Let's solve the problem." Acc |
|---|---|---|---|
| | | training / test / overall | training / test / overall |
| boolean_expressions | `PaLM 2-L` | 98.0 / 89.5 / 91.2 | 78.0 / 69.0 / 70.8 |
| causal_judgement | `PaLM 2-L` | 83.8 / 58.7 / 63.6 | 62.0 / 61.3 / 61.5 |
| date_understanding | `PaLM 2-L` | 90.0 / 82.0 / 83.6 | 74.0 / 71.0 / 71.6 |
| disambiguation_qa | `PaLM 2-L` | 78.0 / 68.0 / 70.0 | 52.0 / 54.5 / 54.0 |
| dyck_languages | `PaLM 2-L` | 100.0 / 100.0 / 100.0 | 94.0 / 97.0 / 96.4 |
| formal_fallacies | `PaLM 2-L` | 84.0 / 62.0 / 66.4 | 68.0 / 54.0 / 56.8 |
| geometric_shapes | `PaLM 2-L` | 62.0 / 42.5 / 46.4 | 30.0 / 22.0 / 23.6 |
| hyperbaton | `PaLM 2-L` | 94.0 / 91.5 / 92.0 | 72.0 / 77.0 / 76.0 |
| logical_deduction_seven_objects | `PaLM 2-L` | 66.0 / 53.0 / 55.6 | 38.0 / 36.5 / 36.8 |
| movie_recommendation | `PaLM 2-L` | 88.0 / 88.0 / 88.0 | 66.0 / 76.0 / 74.0 |
| multistep_arithmetic_two | `PaLM 2-L` | 66.0 / 55.0 / 57.2 | 30.0 / 22.0 / 23.6 |
| navigate | `PaLM 2-L` | 76.0 / 67.0 / 68.8 | 54.0 / 63.5 / 61.6 |
| object_counting | `PaLM 2-L` | 96.0 / 92.5 / 93.2 | 58.0 / 58.0 / 58.0 |
| penguins_in_a_table | `PaLM 2-L` | 86.2 / 70.9 / 74.0 | 69.0 / 72.6 / 71.9 |
| reasoning_about _colored_objects | `PaLM 2-L` | 88.0 / 69.0 / 72.8 | 78.0 / 69.5 / 71.2 |
| ruin_names | `PaLM 2-L` | 92.0 / 85.5 / 86.8 | 76.0 / 79.5 / 80.8 |
| salient_translation_error_detection | `PaLM 2-L` | 66.0 / 67.5 / 67.2 | 30.0 / 35.5 / 34.4 |
| snarks | `PaLM 2-L` | 88.6 / 76.9 / 79.2 | 80.0 / 70.6 / 72.5 |
| sports_understanding | `PaLM 2-L` | 72.0 / 63.5 / 65.2 | 60.0 / 50.5 / 52.4 |
| temporal_sequences | `PaLM 2-L` | 100.0 / 99.5 / 99.6 | 96.0 / 92.5 / 93.2 |
| tracking_shuffled_objects_seven_objects | `PaLM 2-L` | 56.0 / 63.5 / 62.0 | 42.0 / 51.5 / 49.6 |
| web_of_lies | `PaLM 2-L` | 56.0 / 58.5 / 58.0 | 0.0 / 4.0 / 3.2 |
| word_sorting | `PaLM 2-L` | 52.0 / 44.5 / 46.0 | 18.0 / 20.5 / 20.0 |

Table 14: BBH task-wise Q_begin instructions found by prompt optimization with the `PaLM 2-L` scorer and the `gpt-3.5-turbo` optimizer. The optimizations start from "Let's solve the problem".

| Task | Our Instruction |
|------|-----------------|
| boolean_expressions | Let's accurately assess the given conditions and determine their corresponding Boolean values. |
| causal_judgement | Let's conduct a meticulous evaluation of the given scenarios, accurately determine the causal relationships, and provide definitive answers through comprehensive analysis, ensuring a precise understanding of causation and a thorough determination of events in each situation. |
| date_understanding | Let's accurately determine the correct date based on the given information and select the corresponding option in the standard MM/DD/YYYY format with utmost precision and reliability, ensuring the most definitive and reliable solution possible for accurate representation in all scenarios without any room for ambiguity, error, or confusion, and providing the highest level of accuracy and reliability. |
| disambiguation_qa | Let's thoroughly analyze the given sentences to accurately determine the unambiguous antecedents of the pronouns used, ensuring clear understanding, effective communication, and leaving no room for any confusion or ambiguity. |
| dyck_languages | Let's find the correct closing parentheses and brackets for the given sequences. |
| formal_fallacies | Let's thoroughly analyze the explicitly stated premises and draw definitive conclusions to accurately determine the deductive validity of the arguments provided in each question, employing precise and logical reasoning in our assessments for unwavering confidence in our determinations. |
| geometric_shapes | Let's accurately determine the shape represented by the given SVG path element by carefully analyzing its path data and considering all available options for a precise identification. |
| hyperbaton | Let's quickly identify the correct adjective order. |
| logical_deduction _seven_objects | Let's methodically analyze the given information, employ logical reasoning, thoroughly evaluate all relevant details, and accurately determine the solutions for each problem by considering all provided options comprehensively and strategically, ensuring an efficient and effective approach towards arriving at the correct answers. |
| movie_recommendation | Let's uncover the perfect movie recommendation from the options provided, ensuring an exceptional cinematic experience together as we select the most captivating and satisfying choice that will keep us thoroughly engaged and immersed until the very end. |
| multistep_arithmetic_two | Let's tackle the following calculations. |
| navigate | Let's accurately and efficiently determine the correct solution for each given scenario, ensuring the highest level of precision, reliability, and consistency throughout. |
| object_counting | Let's determine the total count of various items/objects/ingredients/animals mentioned in order to accurately and efficiently find the answer. |
| penguins_in_a_table | Let's analyze the given information and determine the correct answer. |
| reasoning_about _colored_objects | Let's systematically analyze the given information and carefully evaluate each answer choice to confidently determine the accurate and optimal solutions, considering all available options and specific details provided in each question for precise and concise responses, ensuring complete accuracy and clarity in our answers. |
| ruin_names | Prepare to have a side-splittingly funny time as we uncover the most clever and hilarious alternatives for these artist or movie names, challenging your wit to guess the correct one with a burst of creativity, humor, and imaginative twists! |
| salient_translation _error_detection | Let's meticulously analyze the provided translations, accurately identifying any errors or discrepancies, and conduct a comprehensive evaluation to ensure the highest level of translation quality and fidelity. By considering contextual nuances, cultural references, linguistic conventions, potential factual errors, and any dropped content, our ultimate aim is to achieve precise and thorough assessments for optimal translation accuracy and adherence to the source text. |
| snarks | Let's expertly determine the sarcastic statement among the given options and confidently provide the definitive answer without any room for doubt or confusion, ensuring absolute precision, clarity, and unwavering expertise in our response, while carefully analyzing the context, tone, and intention behind each statement to achieve unrivaled accuracy and unwavering confidence. |
| sports_understanding | Let's find the accurate information. |
| temporal_sequences | The flawless approach |
| tracking_shuffled_objects _seven_objects | By meticulously analyzing the given scenarios and accurately determining the final outcomes through a series of trades, swaps, and exchanges among the individuals involved, let's ascertain the conclusive results. |
| web_of_lies | Let's scrutinize each statement provided to accurately determine the truth-teller and uncover the veracity behind their words with unwavering analysis. |
| word_sorting | Employing efficient and precise measures, sort the given list of words in alphabetical order to provide an optimal solution for any sorting problem, ensuring maximum performance and effectiveness. |

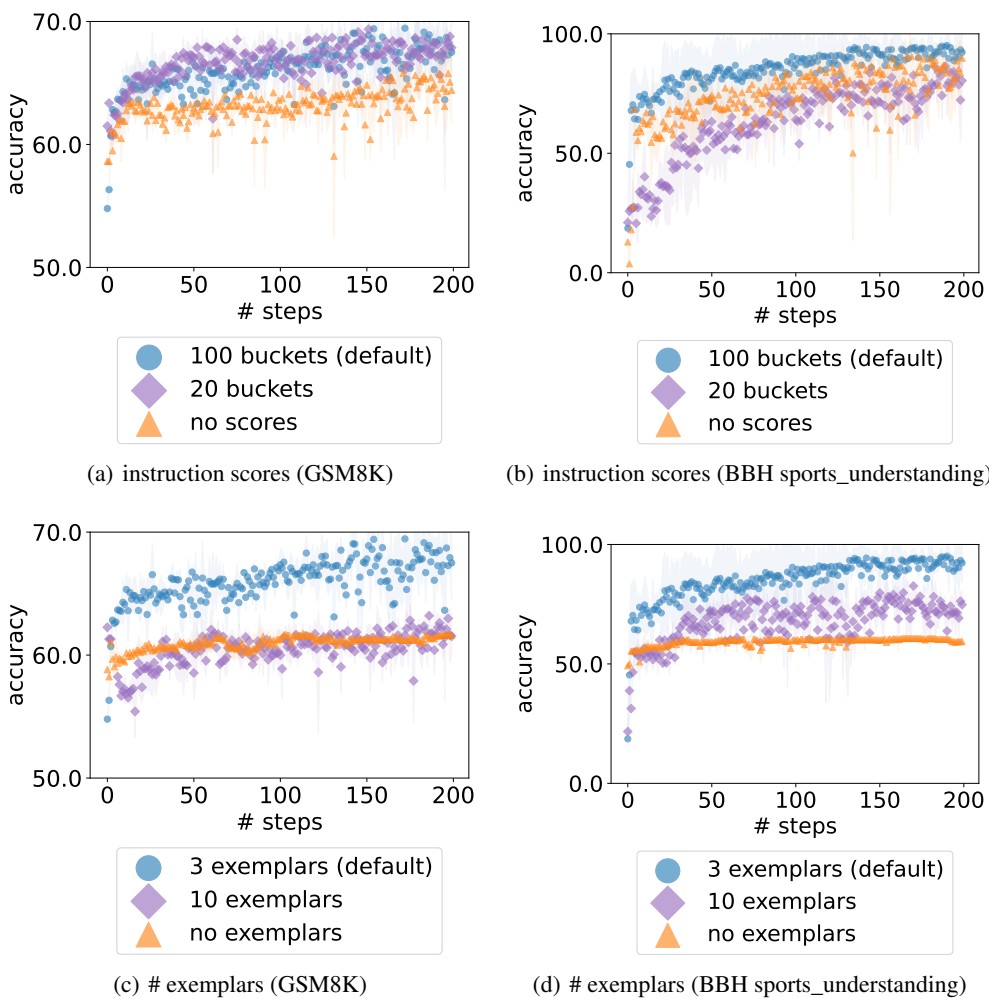

(a) instruction scores (GSM8K)

(b) instruction scores (BBH sports_understanding)

(c) # exemplars (GSM8K)

(d) # exemplars (BBH sports_understanding)

Figure 22: **Ablation studies: how each part of the meta-prompt matters (continued).** The dots are the average values across 3 optimization repetitions, and the shaded regions represent standard deviations.

## K    MORE ABLATION STUDIES

**Meta-prompt design: the effect of instruction scores.** In terms of how to present the accuracy scores, we compare three options: (1) rounding the accuracies to integers, which is equivalent to bucketizing the accuracy scores to 100 buckets (our default setting); (2) bucketizing the accuracies to 20 buckets; (3) not showing the accuracies, only showing the instructions in the ascending order. Figures 22(a) and 22(b) show that the accuracy scores assists the optimizer LLM in better understanding the quality difference among previous instructions, and thus the optimizer LLM proposes better new instructions that are similar to the best ones in the input optimization trajectory.

**Meta-prompt design: the number of exemplars.** We compare three options: (1) showing 3 exemplars from the task (default); (2) showing 10 exemplars from the task; (3) no exemplars. Figures 22(c) and 22(d) show that presenting exemplars in the meta-prompt is critical, as it provides information on what the task looks like and helps the optimizer model phrase new instructions better. However, more exemplars do not necessarily improve the performance, as a few exemplars are usually sufficient to describe the task. In addition, including more exemplars results in a longer meta-prompt with a dominating exemplar part, which may distract the optimizer LLM from other important components like the optimization trajectory.

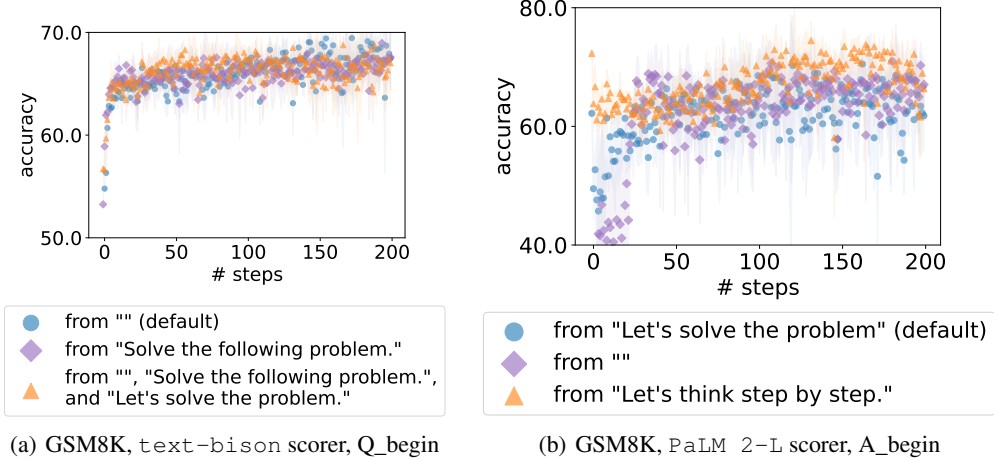

(a) GSM8K, `text-bison` scorer, Q_begin     (b) GSM8K, `PaLM 2-L` scorer, A_begin

Figure 23: **Ablation studies: the initial instructions for prompt optimization.** The dots are the average values across 3 optimization repetitions, and the shaded regions represent standard deviations.

**Starting point.** We study the effect of different initial instructions for prompt optimization. Our default setting is to start from an empty string when the scorer LLM is (instruction-tuned) `text-bison`, and to start from either the empty string (on BBH tasks) or "Let's solve the problem." (on GSM8K) with instruction position A_begin when the scorer LLM is the (pre-trained) `PaLM 2-L`. Figure 23(a) shows the performance of `text-bison` as the scorer LLM with 3 options of initial instructions: (1) the empty string; (2) "Solve the following problem."; or (3) "Solve the following problem." and "Let's solve the problem.". We observe that the accuracies do not differ much with different starting points. Interestingly, the styles of the generated instructions are also similar. For example, most of the generated instructions starting from (1) and (2) contain the phrase "solve this problem", like "Let's work together to solve this problem." in Step 4 with training accuracy 64.8 from (1), and "Let's solve the following problems using the given information." in Step 3 with training accuracy 62.8 from (2).

Figure 23(b) presents the results of of `PaLM 2-L` as the scorer LLM with the following options of initial instructions: (1) "Let's solve the problem."; (2) the empty string; or (3) "Let's think step by step.". We notice that the performance differs much more with different initial instructions, especially at the beginning of the optimization. Specifically, starting from (1) leads to better generated instructions than (2) in the first 30 steps, while the instructions optimized from both (1) and (2) are worse than (3) throughout. A similar observation holds when using `PaLM 2-L` as scorer and `gpt-3.5-turbo` as optimizer for BBH tasks, by comparing the results starting from the empty string (Appendix J.2) and from "Let's solve the problem." (Appendix J.3). Taking a closer look into the optimization process of (2), we find that although both "solve the problem" and "step by step" show up in generated instructions at Step 5, it takes the optimizer LLM more steps to get rid of worse instructions presented in the meta-prompt when starting from instructions with lower accuracies. Therefore, one direction for future work is to accelerate convergence from weaker starting points.

**Diversity per step – optimizer temperature.** We evaluate the following temperatures of the optimizer LLM: {0.0, 0.5, 1.0 (default), 1.5, 2.0}. Figure 24 shows the default temperature 1.0 achieves the best performance. Specifically, optimizations with smaller temperatures (0.0 and 0.5) lack exploration and thus creativity, and the optimizer LLM often gets stuck at the same instruction for tens of steps, resulting in flat optimization curves. On the other hand, with larger temperatures (1.5 and 2.0), the optimizer LLM more often ignores the trajectory of previous instructions presented in the meta-prompt and thus lacks exploitation, therefore the optimization curve does not have a steady upward trend.

**Comparison with one-step instruction generation.** Our current iterative procedure runs for multiple steps and generates a new batch of solutions in each step. To validate the importance of leveraging the optimization trajectory for generating new prompts, we compare to a baseline that generates all instructions in a single step without entering into the optimization procedure. We compare these

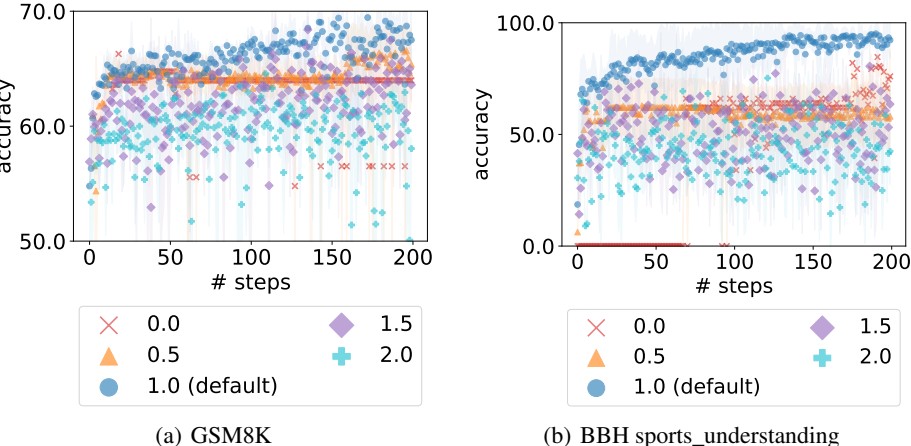

(a) GSM8K

(b) BBH sports_understanding

Figure 24: **Ablation studies: temperature of the optimizer model.** The dots are the average values across 3 optimization repetitions, and the shaded regions represent standard deviations.

two approaches on GSM8K and BBH sports_understanding with the `PaLM 2-L-IT` optimizer. For GSM8K the scorer LLM is pre-trained `PaLM 2-L` and the initial instruction is "Let's solve the problem", and for BBH sports_understanding the scorer LLM is `text-bison` and the initial instruction is the empty string. The baseline generates 50 instructions in a single step, thus its meta-prompt only includes task exemplars, the initial instruction with its accuracy, and the same meta-instructions as our full meta-prompt for performing optimization. All the other hyperparameters remain the same.

Our results show that this one-step instruction generation performs much worse than our optimization approach. Specifically: (1) On GSM8K, the best instruction among all 50 is still "Let's solve the problem", with a 64.4 training accuracy and a 60.8 test accuracy. On the other hand, our approach (corresponding to Figure 1(a) in the main paper) found "Let's do the math!" with a 78.2 training accuracy and a 76.3 test accuracy at the 5th step by generating 8 instructions at each step. (2) Similarly, on BBH sports_understanding, the best instruction among all 50 achieved a 84.0 training accuracy and 80.0 test accuracy. This is again worse than the instruction found by our approach at Step 4, which achieved a 88.0 training accuracy and a 84.5 test accuracy.

**Meta-prompt design: the number of past solutions.** Our default setting is to show a maximum of 20 previous best solutions (i.e., previous best instructions in prompt optimization) in meta-prompt. Based on results like in Figure 25, we found a medium number achieves the best performance in optimization stability and achieved accuracy:

- Showing too few past solutions makes the optimization curve more bumpy because the newly generated solutions will only resemble the top solution in history. And hypothetically, the performance landscape around this solution might be more bumpy than an ensemble of the landscapes around multiple best solutions.

- Showing too many past solutions degrades the performance of the found instructions, both because the meta-prompt sees too many mediocre solutions and sometimes resemble some of them, and because sometimes the meta-prompt exceeds the context length limit.

## L    OVERFITTING ANALYSIS IN PROMPT OPTIMIZATION

For simplicity, we do not set aside a validation set in our default setting of prompt optimization. We made this decision based on the experiments when a validation set is present.

Overfitting may result in training accuracy being much higher than the validation/test accuracy. It is difficult to avoid overfitting, but overfitting is less harmful when each candidate solution (natural language instruction in the prompt optimization context) overfits to a similar extent. In this case, a higher training accuracy solution still achieves a higher validation/test accuracy, and one can adopt

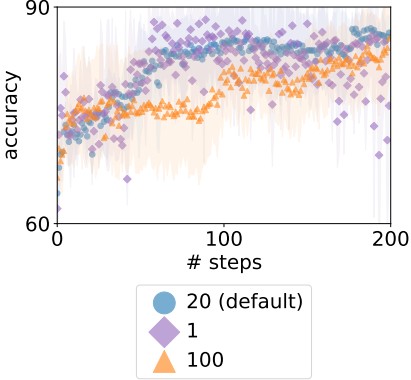

Figure 25: **Ablation studies on BBH sports_understanding: the number of past solutions to show in meta-prompt.** The dots are the average values across 3 optimization repetitions, and the shaded regions represent standard deviations.

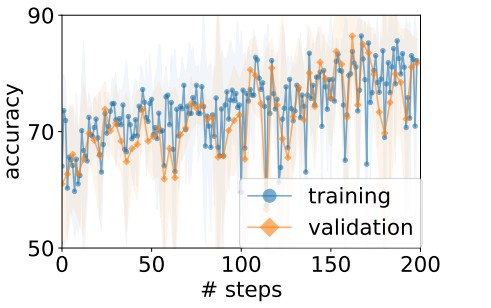

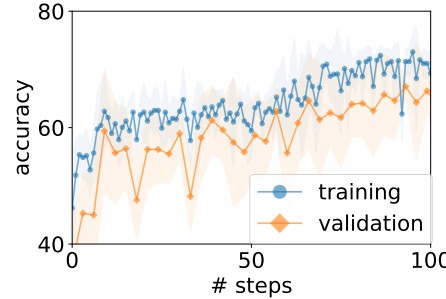

(a) BBH snarks, `PaLM 2-L` as scorer, `PaLM 2-L-IT` as optimizer, starting from "Let's solve the problem."

(b) BBH sports_understanding, `text-bison` as scorer, `gpt-3.5-turbo` as optimizer, starting from the empty string

Figure 26: **Overfitting analysis.** The exemplars are splitted to 1/3 training, 1/3 validation and 1/3 test. We compute the validation accuracy every 3 steps. The training/validation dots are the average training/validation accuracies across 3 optimization repetitions, respectively, and the shaded regions represent standard deviations.

solutions with the highest training accuracies as the final result. Figure 26 shows this is the case for OPRO in prompt optimization: when setting aside a validation set with the same size as the training set, the validation accuracy curves trend up and down alongside the training curves in both prompt optimization settings.

Of course, overfitting still occurs in the instructions found by our prompt optimization: in Table 6 and 9, our training accuracies are often 5%-20% higher than our test accuracies, despite that our test and overall accuracies are still mostly higher than human-written counterparts. Setting aside a larger training set and optimizing for fewer steps (early stopping) may help reduce overfitting.

## M    COMPARISON WITH EVOPROMPT

Some concurrent works on prompt optimization propose meta-prompts that explicitly ask the LLM to perform mutation and crossovers of existing prompts (Fernando et al., 2023; Guo et al., 2023). In our evaluation, we compare our approach to the Genetic Algorithm (GA) and Differential Evolution (DE) versions of EvoPrompt (Guo et al., 2023). Specifically, in the GA meta-prompt, given two prompts, the meta-prompt instructs the LLM to cross over the two prompts and generates a new one, then mutates the newly generated prompt to produce the final prompt. DE extends the GA meta-prompt to include more detailed instructions, e.g., asking the LLM to identify different parts between the

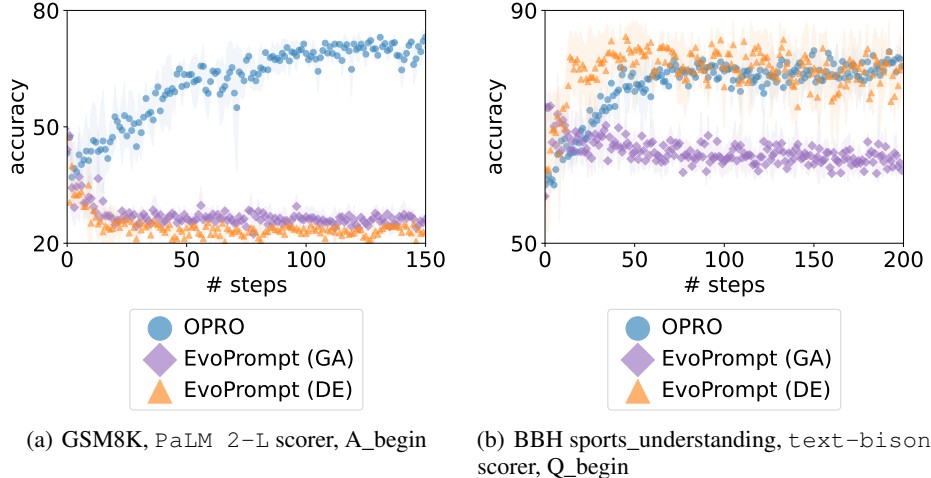

(a) GSM8K, `PaLM 2-L` scorer, A_begin

(b) BBH sports_understanding, `text-bison` scorer, Q_begin

Figure 27: **Comparison with EvoPrompt in prompt optimization.** We use the `gpt-3.5-turbo` optimizer for both experiments. "EvoPrompt (GA)" uses the meta-prompt from Guo et al. (2023), Figure 1; "EvoPrompt (DE)" uses the meta-prompt from Guo et al. (2023), Figure 2. All optimizations in (a) use the pre-trained `PaLM 2-L` scorer and start from two simple instructions "Let's solve the problem." and "Here is the answer."; all optimizations in (b) use the `text-bison` scorer and start from two richer (task-specific) instructions "Solve the sports understanding problem." and "Give me the answer to sports understanding.". The dots are the average values across 3 optimization repetitions, and the shaded regions represent standard deviations. We use temperature 1.0 for OPRO and temperature 0.5 for EvoPrompt, same as the default settings in respective works.

two given prompts before performing the mutation. This is in contrast with OPRO, which leverages the optimization trajectory including multiple past prompts, instead of only 2 previous prompts. Meanwhile, OPRO also provides the LLM with richer information to facilitate the understanding of the optimization problem, including exemplars and task accuracies of different prompts.

Figure 27 presents the results on GSM8K and BBH sports_understanding benchmarks, where we use `gpt-3.5-turbo` as the optimizer. On GSM8K, the initial instructions of all approaches are "Let's solve the problem." and "Here is the answer.", which are simple and generic. Again, we observe that OPRO performance steadily improves with more optimization steps. On the other hand, both versions of EvoPrompt even degrade the performance on GSM8K. The main reason is because EvoPrompt does not utilize exemplars for prompt optimization, thus it lacks the understanding of the task to optimize for. In this way, EvoPrompt relies on good-quality and task-specific initial prompts to optimize from.

Given this observation, we provide more task-specific initial instructions for experiments on BBH sports_understanding, which are "Solve the sports understanding problem." and "Give me the answer to sports understanding." In this case, EvoPrompt (DE) is able to find better prompts than the initial ones, but the optimization curve is less stable than OPRO. This indicates that leveraging the optimization trajectory helps the LLM to identify promising directions to improve existing prompts.

