# OpenReview forum: "Large Language Models as Optimizers"
_ICLR.cc/2024/Conference — ICLR 2024 poster_

### Official Review · Reviewer_iHbx · 2023-10-27

**Soundness:** 3 good
**Presentation:** 3 good
**Contribution:** 2 fair
**Rating:** 8
**Confidence:** 4

**Summary:**

This paper proposes the use of large language models (LLMs) as optimizers to address various optimization tasks, particularly those that can be well expressed in natural language. To validate their approach, the authors conducted experiments on linear regression and traveling salesman problems, as well as prompt optimization, where the goal is to find instructions that maximize task accuracy. Results show that the best prompts optimized by this work outperform human-designed prompts by up to 8% on GSM8K and by up to 50% on Big-Bench Hard tasks. Additionally, the authors evaluated the transferability of found prompts to different datasets in the same domain, demonstrating that their optimized prompts outperform baseline prompts on MultiArith and AQuA.

**Strengths:**

1. This work is among the early investigations into an intriguing research question: can LLMs be used for various optimization tasks?
2. The obtained optimal prompts from the method are both interesting and useful, such as "Take a deep breath and work on this problem step-by-step".
3. The paper writing is clear, and the figures that state the key experiment results and illustrate the method are well-plotted.

**Weaknesses:**

1. What is the key/unique advantage of using LLMs to optimize over some traditional optimization algorithms, especially on classical optimization problems (not prompt engineering)?
2. For prompt optimization, the optimization process of this work (i.e., directly feeding solution-score pairs, optimization task descriptions, and meta-instructions into the optimizer LLM) is a black box/lacks interpretation, why it is a better prompt optimizer than those new methods that leverage LLMs to explicitly act as mutation and crossover operators, and further optimize the prompt? such as,
    1. EvoPrompting: Language Models for Code-Level Neural Architecture Search by Chen et al.
    2. Connecting Large Language Models with Evolutionary Algorithms Yields Powerful Prompt Optimizers by Guo et al.
3. To further understand the effect of the purple text in this work, an ablation study may be beneficial for improving the solidness of the results.

**Questions:**

Please see the above weaknesses

---

> ### Author Response · Authors · 2023-11-18
> **Response to Reviewer iHbx**
>
> Thank you for your careful reading and thoughtful reviews. Let us address your comments below.
>
> > What is the key/unique advantage of using LLMs to optimize over some traditional optimization algorithms, especially on classical optimization problems (not prompt engineering)?
>
> The key/unique advantage is that one can use natural language to describe the optimization problem, as discussed in the second paragraph of Section 1: “​​instead of formally defining the optimization problem and deriving the update step with a programmed solver”. This can make optimization more accessible to general users who do not have much domain knowledge of the types of optimization tasks in question, and may also boost productivity of optimization experts who work on these tasks on a daily basis.
>
> Meanwhile, we would like to clarify that the focus of our work is not to solve all classic optimization problems. We evaluated our approach on linear regression and Traveling Salesman Problems only as motivating examples to demonstrate the potential of LLMs for optimization. Instead, our main task is prompt optimization which is a known challenge for classic optimization methods, where the optimization space is in natural language.
>
> > For prompt optimization, the optimization process of this work (i.e., directly feeding solution-score pairs, optimization task descriptions, and meta-instructions into the optimizer LLM) is a black box/lacks interpretation, why it is a better prompt optimizer than those new methods that leverage LLMs to explicitly act as mutation and crossover operators, and further optimize the prompt?...
>
> Thanks for pointing out these related works. We have discussed such evolutionary methods in Section 6. Following your suggestion, we also added an empirical comparison with EvoPrompt. **Please refer to our general response for a summary, and we summarized the full results in Appendix M of the updated Supplementary Material.**
>
> > To further understand the effect of the purple text in this work, an ablation study may be beneficial for improving the solidness of the results.
>
> We have benchmarked the effect of exemplars (i.e., the purple text) in ablation studies. Please refer to Figure A17 in Appendix K of the Supplementary Material. We show that adding exemplars boosts performance, since it offers more context to the optimizer LLM and enables it to generate more relevant instructions (prompts). For example, on BBH tasks, the optimizer LLM found instructions that are tailored to the tasks, as shown in Table A4 and A5 in Supplementary Material. Without exemplars in the meta-prompt, it is impossible for the optimizer LLM to understand the task if the initial prompts are simple and generic.

---

> > ### Author Response · Authors · 2023-11-21
> > **Follow-up**
> >
> > Thanks again for reviewing our work! Please let us know if we have adequately addressed your questions and concerns.

---

### Official Review · Reviewer_hsEq · 2023-10-30

**Soundness:** 3 good
**Presentation:** 3 good
**Contribution:** 3 good
**Rating:** 8
**Confidence:** 4

**Summary:**

This paper proposes to use LLMs as optimizers by simply inputting the natural language description of the optimization task, previous steps’ inputs and scores to LLMs. This paper applies such method on prompt search for various LLM tasks and demonstrates its effectiveness.

**Strengths:**

1. Good proof of concept. This paper provides concrete evidence that large language models can find the patterns between the inputs and corresponding scores that humans might not be able to find to conduct optimization tasks.
2. Good use case. Based on such proof of concept, this paper finds a valid use case for the proposed method, on which other traditional optimizations might be difficult to apply, finding the proper prompts for LLM tasks.
3. Solid experiments on prompt search. Experiments show that it is not random that the proposed method is able to find proper prompts for various tasks which lead to significant performance improvements. The thorough ablations also already provides answers to a lot of concerns.

**Weaknesses:**

Two questions on the ablation study.

1. Numbers of examplers. Did you take the randomness of example picking into consideration? For each run of every setting, do you give the same set of examples?
2. I noticed that for different tasks, the “batch size” that works the best can be different (Figure 5, cd). Do you find any obvious patterns on which types of data/tasks prefer a smaller “batch size” and vice versa?

**Questions:**

See the weakness.

---

> ### Author Response · Authors · 2023-11-18
> **Response to Reviewer hsEq**
>
> Thank you for your careful reading and thoughtful reviews. Let us address your comments below.
>
> > Numbers of examplers. Did you take the randomness of example picking into consideration? For each run of every setting, do you give the same set of examples?
>
> At each step, we randomly sample a different set of 3 exemplars to put into the meta-prompt.
>
> - How we chose 3 for the number of exemplars: Our ablation (Figure A17(c) and A17(d) in Appendix K of the Supplementary Material) shows that 3 achieves a good balance between the informativeness of the meta-prompt and the context length constraint.
> - How randomness is useful: We randomly sample exemplars at each step because it reduces the risk of overfitting on a fixed set of exemplars, and reduces the chance of the optimization getting stuck. The latter is because if both the set of exemplars and the top instructions in the meta-prompt remain the same (this more often occurs in later steps when the performance starts to saturate), the meta-prompt will remain the same, and the optimizer LLM may keep generating the same instructions.
>
> > I noticed that for different tasks, the “batch size” that works the best can be different (Figure 5, cd). Do you find any obvious patterns on which types of data/tasks prefer a smaller “batch size” and vice versa?
>
> Insightful question. Although setting the batch size of 8 generally achieves the best final performance across models and tasks, some other batch sizes may work equally well in certain cases (like batch size 1 and 2 at the early stage of optimization in Figure 5(d)). In general, the batch size used by our approach can be smaller for easier optimization problems, such as:
> - When the scorer LLM is instruction-tuned, the accuracy landscape will be less bumpy because the model performance is less sensitive to the prompt, thus a smaller batch size would suffice for the accuracy curve to trend up. The bumpiness can be well reflected by the oscillation in the optimization curve: both on the same GSM8K 3.5% training set, the curve on the pre-trained PaLM 2-L scorer (Figure 1(a)) is much more bumpy than the blue dots in Figure 5(a), especially at early steps. This means OPRO on a pre-trained scorer LLM needs a larger batch size (like 8) to succeed.
> - When the task is easier, one doesn’t need to carefully prompt the model to get a good score, meaning the loss landscape is also less bumpy. One example is the difference between Figure 5(c) and 5(d) in the main paper: BBH sports_understanding is easier than GSM8K in that the same text-bison scorer LLM achieves around 30% higher accuracy, close to 100% on sports_understanding, thus a smaller batch size is enough.

---

> > ### Author Response · Authors · 2023-11-21
> > **Follow-up**
> >
> > Thanks again for reviewing our work! Please let us know if we have adequately addressed your questions and concerns.

---

### Official Review · Reviewer_3F2p · 2023-11-01

**Soundness:** 3 good
**Presentation:** 3 good
**Contribution:** 3 good
**Rating:** 6
**Confidence:** 4

**Summary:**

This paper proposes Optimization by Prompting (OPRO), a method to use large language models (LLMs) as optimizers for various tasks. The key idea is to describe the optimization problem and provide the model with past solution-score pairs in a meta-prompt. The LLM then generates new candidate solutions based on this context. OPRO is first demonstrated on linear regression and traveling salesman problems. The main application is prompt optimization, where the goal is to find an instructional prompt that maximizes a model's accuracy on a dataset. Experiments optimize prompts for GSM8K and BigBench, starting from poor prompts and showing significant gains.

**Strengths:**

- Novel idea of leveraging LLMs' understanding of natural language and few-shot learning abilities for optimization. Enables optimization by simply describing the problem rather than formal specification.

- Demonstrated on diverse tasks - mathematical optimization, prompt optimization. Shows potential breadth of this approach.

- Compelling results on prompt optimization. Optimized prompts substantially outperform human-written prompts, improving accuracy by up to 8% on GSM8K and 50% on BigBench.

- Principled design of the meta-prompt, balancing past solution-score pairs and problem description. Ablations validate design choices.

- Thorough experiments comparing different LLMs, scoring models, prompt positions, and baselines. Shows consistent benefits of OPRO.

**Weaknesses:**

- The biggest limitation is that OPRO's performance looks highly fluctuating. It's unclear if the LLM really finds the so-called optimization "trajectory" or just randomly finds a good prompt. The authors should provide more analysis to show that the LLM is indeed learning to optimize.

- Limited exploration on how to provide richer feedback to LLM beyond aggregated scores. It could help address limitations.

- Unclear how sensitive results are to meta-prompt design and hyperparameters like temperature.

- No comparison to other prompt optimization methods. It could better situate contributions.

- Limited analysis. For example, there is no characterization of what makes an effective prompt.

**Questions:**

- Can you clearly state how many optimization steps are performed in each experiment? How does the number of steps affect performance?

- Can you provide an experiment where you generate the same number of prompts in one step as your current experiments, and evaluate them all, and report the best one? This would help clarify whether the LLM is really learning to optimize or just randomly finding a good prompt.

- For prompt optimization, have you experimented with providing more detailed feedback to the LLM beyond aggregated scores? (e.g. accuracy on different example types, common mistakes)

- How does the meta-prompt length affect optimization performance? Is there a sweet spot balancing past solutions and problem descriptions?

- What determines the choices of sampling temperature? Have you tried adaptive temperature schedules?

- What are the limitations on problem complexity that OPRO can handle? Analysis of how performance degrades with complexity?

- Can you better characterize what makes an effective prompt for optimization? Any semantic or syntactic patterns?

- How does OPRO compare to other gradient-free prompt optimization methods? Could be included in experiments.

- Is there any overfitting during prompt optimization? How does test accuracy compare to training accuracy?

---

> ### Author Response · Authors · 2023-11-18
> **Response to Reviewer 3F2p - Part 1/3**
>
> Thank you for your careful reading and thoughtful reviews. Let us address your comments below.
>
> > The biggest limitation is that OPRO's performance looks highly fluctuating. It's unclear if the LLM really finds the so-called optimization "trajectory" or just randomly finds a good prompt. The authors should provide more analysis to show that the LLM is indeed learning to optimize.
>
> We acknowledge that the optimization curves do not show monotonic increase at every step. Most optimizers don’t, especially when the loss landscape has a more complicated curvature. From our new experiment showing that the one-step prompt generation method you suggested performs much worse than our approach, we observe that LLM gradually improves upon previous prompts based on the optimization trajectory, just like what other optimizers do.
>
> In addition, in figures showing the accuracy at each step (Figure 1 in main paper; Figures A11, A12, A13 in Supplementary Material), we presented the *average* (instead of the maximum) accuracy of 8 instructions generated at each step, which showed generally upward trends. This shows the optimizer LLM gradually generates *distributionally* better instructions. Furthermore, Figure 5(c) and 5(d) showed that with the same number of total evaluated instructions, generating 8 instructions per step (default setting) achieved better performance than generating 16 instructions per step, demonstrating that the LLM learns to optimize prompts from previous attempts better with more iterations of update, instead of just randomly finding a good prompt by sampling more instructions.
>
> > Limited exploration on how to provide richer feedback to LLM beyond aggregated scores. It could help address limitations...For prompt optimization, have you experimented with providing more detailed feedback to the LLM beyond aggregated scores? (e.g. accuracy on different example types, common mistakes)
>
> As noted in the discussion of future work in Section 7 (Conclusion), “we tried including error cases in the meta-prompt rather than randomly sampling from the training set at each optimization step, but the results are similar, indicating that the error cases alone are not informative enough for the optimizer LLM to grasp the cause of the wrong prediction”. As also noted in Section 7, we agree that improving our approach to better incorporate more detailed feedback is a promising direction for future work.
>
> That said, we show that our simple meta-prompt still demonstrates significant improvement across different tasks and LLMs, and we see simplicity as a key advantage of our method.
>
> > Unclear how sensitive results are to meta-prompt design and hyperparameters like temperature.
>
> Please see our responses to your respective questions in [Part 2/3](https://openreview.net/forum?id=Bb4VGOWELI&noteId=BRP1leKDJZ).
>
> > Limited analysis. For example, there is no characterization of what makes an effective prompt.
>
> We have studied both aspects in the ablation studies. Please see Section 5.3 in the main paper and Appendix K in the Supplementary Material for ablations on how different parts of the meta-prompt and temperature matter.
>
> > No comparison to other prompt optimization methods. It could better situate contributions.
>
> Thanks for your suggestion. We added an empirical comparison to EvoPrompt, which is an evolutionary method pointed out by Reviewer iHbx. **Please refer to our general response for a summary, and we summarized the full results in Appendix M of the updated Supplementary Material.**
>
> > Can you clearly state how many optimization steps are performed in each experiment? How does the number of steps affect performance?
>
> We run a maximum of 200 steps in every experiment although many converge much sooner. The training accuracies trend upwards until plateaued, so the performance generally improves with more number of steps. As shown in Figure 1 in the main paper and Figures A12, A13 in the Supplementary Material, the performance often plateaus within 100 steps.
>
> **(to be continued in Part 2/3)**

---

> > ### Author Response · Authors · 2023-11-18
> > **Response to Reviewer 3F2p - Part 2/3**
> >
> > **(following Part 1/3)**
> >
> > > Can you provide an experiment where you generate the same number of prompts in one step as your current experiments, and evaluate them all, and report the best one? This would help clarify whether the LLM is really learning to optimize or just randomly finding a good prompt.
> >
> > Sure, based on your suggestion, we conducted the following additional experiment on GSM8K and BBH sports_understanding with the PaLM 2-L-IT optimizer. For GSM8K the scorer LLM is pre-trained PaLM 2-L and the initial instruction (i.e. prompt) is “Let’s solve the problem”, and for BBH sports_understanding the scorer LLM is text-bison and the initial instruction is the empty string. We generate 50 instructions (i.e. prompts) in only one step, thus the meta-prompt only includes task exemplars, the performance of the empty string, and some meta-instructions. All the other hyperparameters are the same as our current experiments in paper.
> >
> > Our results show that this one-step prompt generation performs much worse than our approach. Specifically:
> >
> > - On GSM8K, the best instruction among all 50 is still “Let’s solve the problem”, with a 64.4 training accuracy and a 60.8 test accuracy. On the other hand, our approach (corresponding to Figure 1(a) in the main paper) found “Let’s do the math!”  with a 78.2 training accuracy and a 76.3 test accuracy at the 5th step by generating 8 instructions at each step.
> > - Similarly, on BBH sports_understanding, the best instruction among all 50 achieved a 84.0 training accuracy and 80.0 test accuracy. This is again worse than the instruction found by our approach at Step 4, which achieved a 88.0 training accuracy and a 84.5 test accuracy.
> >
> > **We have also added the above results to a new paragraph “Comparison with one-step instruction generation” in Appendix K of the Supplementary Material.**
> >
> > > How does the meta-prompt length affect optimization performance? Is there a sweet spot balancing past solutions and problem descriptions?
> >
> > As for the meta-prompt length: we have benchmarked the effect of the number of exemplars in Appendix K of the Supplementary Material, where we show that while including exemplars is crucial for achieving good optimization performance, including 10 exemplars in meta-prompt harms the optimization performance, which might be because the total length of exemplars becomes too long that distracts the attention to other important components such as past solutions.
> >
> > Regarding the number of past solutions: we observe that the optimization performance is not sensitive to the number of past solutions in the meta-prompt, as long as that number is not too  small (so that the generated new solution may lack diversity) or too large (exceeding the context length limit). So we include a maximum of 20 top solutions from the past by default. **Please refer to the paragraph “Meta-prompt: the number of past solutions” in Appendix K of the updated Supplementary Material.**
> >
> > > What determines the choices of sampling temperature? Have you tried adaptive temperature schedules?
> >
> > We have an ablation study on sampling temperature in Appendix K of the Supplementary Material (Figure A19). The temperature cannot be too small (so the generated solutions lack diversity) or too large (so the generated solutions do not incorporate enough information from the meta-prompt). We tried linearly increasing or decreasing the temperature throughout the optimization process, but the performance didn’t improve on either GSM8K or BBH, so we stick to a constant temperature.
> >
> > > What are the limitations on problem complexity that OPRO can handle? Analysis of how performance degrades with complexity?
> >
> > OPRO is able to handle all prompt optimization tasks we have tried. For math optimization, we discuss some failure cases at the end of Section 3 in the main paper (for the Traveling Salesman Problem) and in Appendix C of the Supplementary Material. The main takeaways are:
> >
> > - OPRO may “get stuck at a point that is neither global nor local optimal”. This happens more frequently when the problem’s dimensionality gets higher, because there are more coordinates to balance in between. In fact, OPRO often only updates one dimension in each step (a.k.a. coordinate descent), which naturally fails more often on higher dimensionality problems.
> > - OPRO is “hard to navigate a bumpy loss landscape”, for example, the Rosenbrock function.
> > - On the Traveling Salesman Problem, “the performance of OPRO degrades dramatically on problems with larger sizes”.
> >
> > > Can you better characterize what makes an effective prompt for optimization? Any semantic or syntactic patterns?
> >
> > The last paragraph in Section 4 of the main paper details the crucial components of the meta-prompt, and our ablation studies in Section 5.3 of the main paper and Appendix K of the Supplementary Material Section K demonstrate the necessity and design choices of these components.
> >
> > **(to be continued in Part 3/3)**

---

> ### Author Response · Authors · 2023-11-18
> **Response to Reviewer 3F2p - Part 3/3**
>
> **(following Part 2/3)**
>
> > How does OPRO compare to other gradient-free prompt optimization methods? Could be included in experiments.
>
> We added an empirical comparison to EvoPrompt [1], an evolutionary method pointed out by Reviewer iHbx. **Please refer to our general response for a summary, and we summarized the full results in Appendix M of the updated Supplementary Material.**
>
> > Is there any overfitting during prompt optimization? How does test accuracy compare to training accuracy?
>
> Great question, we’ve studied overfitting at an early stage of this work, and the results from there influenced our final approach. **We added Appendix L in the Supplementary Material with training and validation accuracy curves.** To summarize, we observe that the trends of training and validation accuracies are consistent with each other throughout the optimization.
>
> References:
>
> [1] Qingyan Guo, Rui Wang, Junliang Guo, Bei Li, Kaitao Song, Xu Tan, Guoqing Liu, Jiang Bian, Yujiu Yang. Connecting Large Language Models with Evolutionary Algorithms Yields Powerful Prompt Optimizers. arXiv:2309.08532.

---

> > ### Author Response · Authors · 2023-11-21
> > **Follow-up**
> >
> > Thanks again for reviewing our work! Please let us know if we have adequately addressed your questions and concerns.

---

> > > ### Comment · Reviewer_3F2p · 2023-11-21
> > > **Thank you for the response!**
> > >
> > > Thank the authors for the detailed responses. My concerns have been well addressed, and I have raised my recommendation for acceptance. Congratulations on your work!
> > >
> > > Reviewer 3F2p

---

> > > > ### Author Response · Authors · 2023-11-21
> > > > **Thanks**
> > > >
> > > > Thanks Reviewer 3F2p for the prompt reply and the recognition!

---

### Official Review · Reviewer_um42 · 2023-11-01

**Soundness:** 2 fair
**Presentation:** 2 fair
**Contribution:** 2 fair
**Rating:** 5
**Confidence:** 4

**Summary:**

The authors propose large language models (LLMs) as optimizers for various tasks by instructing the models through natural language prompts. Currently, optimization problems have to be explicitly defined, and algorithms are tailored and fine-tuned for specific tasks, which can be challenging and time-consuming. This approach proposes Optimization by PROmpting (OPRO), which leverages the adaptability and versatility of LLMs by modifying the problem description in the prompt, enabling simple and effective optimization for different tasks.
The significant result is that OPRO can lead to better performance on selected language tasks, outperforming human-designed prompts by up to 8% on GSM8K and 50% on Big-Bench Hard tasks.

**Strengths:**

This work demonstrates that LLMs can help optimize prompts to achieve high performance on a variety of tasks.

**Weaknesses:**

First, I disagree with the authors fundamentally about what optimization means. To me, this work is not optimization but step-by-step inference. To quote Wikipedia for reference, optimization is "the selection of a best element, with regard to some criterion, from some set of available alternatives." One can plausibly consider the process of prompt selection as "optimization", but in order to make a claim on the general area of optimization I would expect results on optimizing a wide range of convex functions and non convex functions as opposed to word problems. The claim on linear regression as an important result is a relevant but very limited result, but it is not included in the main paper.

Second, considering the relevance of this approach to step-by-step inference, what is new here compared to previous step-by-step inference procedures? The authors include them in the introduction as evidence for LLM is capable of doing multi-step reasoning, but do not distinguish differences b/w this work on prior work. Also they do not compare to these prior work.

**Questions:**

What is the difference between this work and other step-by-step inference techniques such as https://arxiv.org/pdf/2205.11916, https://arxiv.org/abs/2201.11903, https://arxiv.org/abs/2305.10601

How do the results compare?

---

> ### Author Response · Authors · 2023-11-18
> **Response to Reviewer um42**
>
> Thank you for your careful reading and thoughtful reviews. Let us address your comments below.
>
> > First, I disagree with the authors fundamentally about what optimization means. To me, this work is not optimization but step-by-step inference...
>
> First, we would like to clarify that the focus of our work is not to solve all kinds of classic optimization problems. We evaluated our approach on linear regression and traveling salesman problems only as motivating examples to demonstrate the potential of LLMs for optimization. Instead, our main task is prompt optimization which is a known challenge for classic optimization methods, where the optimization space is in natural language.
>
> Meanwhile, optimization is a very general concept. As classic optimizers such as gradient descent and Adam, our approach proposes better solutions in an iterative manner, step-by-step. Specifically, instead of updating the current solution with a mathematical rule (like in momentum or Adam) or by inferencing a surrogate model (like in Bayesian optimization), we prompt the LLM to get the next-step solution, hence “LLMs as optimizers”. From this perspective, prompt optimization is also an optimization problem: the objective is to maximize the task accuracy within the huge natural language space.
>
> > Second, considering the relevance of this approach to step-by-step inference, what is new here compared to previous step-by-step inference procedures?...What is the difference between this work and other step-by-step inference techniques such as https://arxiv.org/pdf/2205.11916, https://arxiv.org/abs/2201.11903, https://arxiv.org/abs/2305.10601 How do the results compare?
>
> These works are complementary to our work. All 3 works used *manually written prompts* to improve LLMs’ performance, while our method focuses on *how to automatically find such prompts via optimization* and in many cases we can find even better prompts compared to the manually written ones in these existing works.
>
> More specifically, https://arxiv.org/pdf/2205.11916 shows LLMs can do zero-shot chain-of-thought prompting; https://arxiv.org/abs/2201.11903 shows chain-of-thought prompting improves LLMs’ reasoning performance; https://arxiv.org/abs/2305.10601 shows tree-of-thought improves the performance on several tasks that require planning and search. These papers are about manually improving the LLM performance with manually written prompts, while our approach is on prompt optimization/tuning. For example, our work showed how our method greatly improved zero-shot chain-of-thought prompting.

---

> > ### Comment · Reviewer_um42 · 2023-11-18
> > **Response**
> >
> > > First, we would like to clarify that the focus of our work is not to solve all kinds of classic optimization problems. We evaluated our approach on linear regression and traveling salesman problems only as motivating examples to demonstrate the potential of LLMs for optimization. Instead, our main task is prompt optimization which is a known challenge for classic optimization methods, where the optimization space is in natural language.
> >
> > I agree with your qualification. But the manuscript is written to present the contribution as one to optimization, not prompt optimization. Both the abstract and the introduction lead with optimization as a motivation.
> >
> > > Meanwhile, optimization is a very general concept. As classic optimizers such as gradient descent and Adam, our approach proposes better solutions in an iterative manner, step-by-step. Specifically, instead of updating the current solution with a mathematical rule (like in momentum or Adam) or by inferencing a surrogate model (like in Bayesian optimization), we prompt the LLM to get the next-step solution, hence “LLMs as optimizers”. From this perspective, prompt optimization is also an optimization problem: the objective is to maximize the task accuracy within the huge natural language space.
> >
> > Prompt optimization is one optimization problem, but showing results for one optimization problem does not make a convincing argument for a general optimization contribution. Like the other review said, this work does not show evidence on, say, classical optimization tasks.
> >
> > > These works are complementary to our work. All 3 works used manually written prompts to improve LLMs’ performance, while our method focuses on how to automatically find such prompts via optimization and in many cases we can find even better prompts compared to the manually written ones in these existing works.
> >
> > These techniques are methods to automatically create prompts, not manually written prompts. For instance, copying execution outputs into the prompt (e.g. ReAct) requires no "manual writing". I would argue that the proposed method is very similar to copying execution outputs (in this case solution-score pairs) into the input.

---

> > > ### Author Response · Authors · 2023-11-18
> > > **Response to Reviewer um42's further response - Part 1/2**
> > >
> > > > I agree with your qualification. But the manuscript is written to present the contribution as one to optimization, not prompt optimization. Both the abstract and the introduction lead with optimization as a motivation.
> > >
> > > Again, we would like to reiterate that our work proposes to leverage LLMs for optimization tasks that are described in natural language, where prompt optimization is a representative application. Our results on linear regression (continuous optimization), traveling salesman problems (combinatorial optimization) and prompt optimization (a hard discrete optimization problem where classic optimizers are inapplicable) demonstrate that LLMs can perform iterative optimization for diverse tasks.
> > >
> > > On the other hand, as stated in Section 3 and Appendices B and C, we did not claim that LLMs can solve every optimization problem and outperform classic optimization algorithms specialized for certain tasks. Instead, we provided a thorough discussion on success and failure cases of LLMs for solving classic optimization problems, which enables a more comprehensive understanding of LLM capabilities as optimizers.
> > >
> > > > Prompt optimization is one optimization problem, but showing results for one optimization problem does not make a convincing argument for a general optimization contribution. Like the other review said, this work does not show evidence on, say, classical optimization tasks.
> > >
> > > As discussed above, we’ve shown results on three optimization tasks that span continuous optimization, combinatorial optimization, and prompt optimization (a hard discrete optimization problem where classic optimizers are inapplicable). Our results demonstrate that LLMs can perform iterative optimization for diverse tasks. Furthermore, as stated in Section 3 and Appendices B and C, we did not claim that LLMs can solve every optimization problem and outperform classic optimization algorithms specialized for certain tasks. Instead, we provided a thorough discussion on success and failure cases of LLMs for solving classic optimization problems, which enables a more comprehensive understanding of LLM capabilities as optimizers.
> > >
> > > Regarding other reviews, we would like to point out that [Reviewer 3F2p](https://openreview.net/forum?id=Bb4VGOWELI&noteId=fLCRDq9IjJ) described “Demonstrated on diverse tasks - mathematical optimization, prompt optimization. Shows potential breadth of this approach” as a strength of our work, and [Reviewer iHbx](https://openreview.net/forum?id=Bb4VGOWELI&noteId=lEn6qgvQgY) acknowledged that “the authors conducted experiments on linear regression and traveling salesman problems, as well as prompt optimization” and our work “is among the early investigations into an intriguing research question: can LLMs be used for various optimization tasks?”.
> > >
> > > **(to be continued in Part 2/2)**

---

> > > > ### Author Response · Authors · 2023-11-18
> > > > **Response to Reviewer um42's further response - Part 2/2**
> > > >
> > > > **(following Part 1/2)**
> > > >
> > > > > These techniques are methods to automatically create prompts, not manually written prompts. For instance, copying execution outputs into the prompt (e.g. ReAct) requires no "manual writing". I would argue that the proposed method is very similar to copying execution outputs (in this case solution-score pairs) into the input.
> > > >
> > > > Again, we would like to clarify that the works you mentioned are all about manually written prompts. Specifically:
> > > > - Chain-of-thought prompting paper (https://arxiv.org/abs/2201.11903) repetitively stated that they “manually composed chains of thought to use as few-shot exemplars” in sections on experiments (Section 3-5).
> > > > - Zero-shot chain-of-thought prompting paper (https://arxiv.org/pdf/2205.11916) manually designed zero-shot instructions to trigger chain of thought, and show that “Let’s think step by step” achieves the best performance among all their manually written instructions.
> > > > - Tree-of-thought prompting paper (https://arxiv.org/abs/2305.10601) manually designed prompts for thought generation and thought evaluation.
> > > > - ReAct (https://arxiv.org/pdf/2210.03629.pdf) stated that its prompting scheme “manually compose ReAct-format trajectories to use as few-shot exemplars in the prompts” (e.g., in Section 3.2).
> > > >
> > > > In our evaluation on prompt optimization, our focus is to show that LLMs can automatically create zero-shot prompts, and optimize over manually written prompts. As presented in Table 1 of the paper, we demonstrated that the best prompt automatically produced by our approach achieved 80.2% accuracy on GSM8K with PaLM 2-L, which significantly outperforms zero-shot chain-of-thought prompting with “Let’s think step by step” (71.8%), and matches the few-shot chain-of-thought prompting performance (80.7%, as shown in PaLM-2 Technical Report https://arxiv.org/pdf/2305.10403.pdf, Page 15 Table 7), where the few-shot chain-of-thought prompt was manually written.
> > > >
> > > > The process of “copying execution outputs into the prompt” is part of a wide variety of approaches, including ReAct, LLMs for tool use (e.g., https://arxiv.org/abs/2302.04761), LLMs for code generation (e.g., https://arxiv.org/abs/2304.05128), etc. Including this process itself does not degrade the novelty of our work. Instead, our core contribution is to show that LLMs can learn to optimize from the trajectory of solution-score pairs, which is fundamentally different from ReAct that utilizes LLMs to interact with the knowledge base or an embodied environment for decision making.

---

> > > > > ### Comment · Reviewer_um42 · 2023-11-18
> > > > > **Response**
> > > > >
> > > > > Thank you. I believe I had fundamentally misunderstood how the metaprompt is created in this work. After reading the authors' description of few-shot prompt curation and zero-shot prompting, I do think there is some distinction between this and the ReAct body of work. That said, I think the degree of "manual prompting" between "let's think step by step" (e.g. zero-shot chain-of-thought) and "increase the score" (e.g. this work) is the same. Finally, given the relevance of these works, I think the authors should explore the similarities and differences between the proposed method and these works in the manuscript.
> > > > >
> > > > > I will increase my score. However, I still take issue with the paper's presentation and framing, which poses the proposed method as a general-purpose optimization method. If this is the path the authors intend to go, then I would suggest putting the linear regression results into the main body, and presenting some more difficult optimization problems (e.g. one with more dimensions and nonlinearity such as MLP on MNIST).

---

> > > > > > ### Author Response · Authors · 2023-11-19
> > > > > > **Response to Reviewer um42's 2nd further response**
> > > > > >
> > > > > > Thank you for revisiting the understanding of our work and increasing the score! Let us further address your comments.
> > > > > >
> > > > > > > That said, I think the degree of "manual prompting" between "let's think step by step" (e.g. zero-shot chain-of-thought) and "increase the score" (e.g. this work) is the same. Finally, given the relevance of these works, I think the authors should explore the similarities and differences between the proposed method and these works in the manuscript.
> > > > > >
> > > > > > We agree that our meta-prompt was manually designed. However, as discussed in our previous responses and in the paper, the main difference between our work and existing work on manual prompting is that our approach automatically discovers better prompts than manually-written ones across tasks, where the manual design of the meta-prompt and the optimization scheme is a one-time effort. Specifically:
> > > > > >
> > > > > > - The design of the meta-prompt and the optimization scheme requires less manual effort. It is notoriously difficult to manually tune prompts for diverse tasks and LLMs, because LLMs are shown to be sensitive to the prompt format (the 4th paragraph in Section 1 Introduction), and semantically similar prompts may have drastically different performance (Section H in Supplementary Material). On the other hand, the meta-prompt design follows the same structure (Section 2.2 in the main paper) and is directly applicable across tasks. As shown in our evaluation, the high-performance prompts found by our approach are diverse for different tasks (Table 1 in main paper, Table A4, A5, A7, A8 in the Supplementary Material), and it would be extremely hard and time-consuming for humans to come up with them.
> > > > > > - With the shared design of the meta-prompt and the optimization scheme, our approach automatically found diverse prompts that work better than manually designed prompts on GSM8K (Table 1 in main paper), MultiArith, AQuA (Table 3 in main paper), and BBH tasks (Table A3, A6, A10 in the Supplementary Material).
> > > > > >
> > > > > > > I would suggest putting the linear regression results into the main body
> > > > > >
> > > > > > Thanks for your suggestion. We agree that linear regression is an integral part of our experiments, and will definitely put these results in the main paper in our revised version. Currently we put the linear regression experiments in the appendix due to the space limit, as we would like to highlight our main results on prompt optimization in the main paper. However, we presented the results on the traveling salesman problem in the main paper (Section 3), which is a long-standing classic optimization task where finding the optimal solution is much more challenging than linear regression.
> > > > > >
> > > > > > > and presenting some more difficult optimization problems (e.g. one with more dimensions and nonlinearity such as MLP on MNIST)
> > > > > >
> > > > > > We acknowledge that currently our approach can “get stuck at a point that is neither global nor local optimal” (Appendix C in the Supplementary Material). This happens more frequently when the problem’s dimensionality gets higher, because there are more coordinates to balance in between. For optimizing MLP parameters on MNIST, another practical concern is the context length limit. Specifically, the MNIST training set includes 60K images of size 28x28, along with presenting the MLP structure with all parameters, fitting the optimization problem description into the LLM context itself becomes a challenge. We consider addressing these issues as future work (for example, with long-context and multimodal models).

---

> ### Author Response · Authors · 2023-11-21
> **Follow-up**
>
> Thanks again for your review and your follow-up discussion! Please let us know if we have adequately addressed your questions and concerns.

---

### Author Response · Authors · 2023-11-18
**Response to All Reviewers**

We thank the reviewers for their constructive comments. We appreciate the reviewers’ positive feedback that our approach is novel (3F2p, iHbx) with thorough experiments (3F2p, hsEq) and compelling results on diverse tasks (3F2p, hsEq). We have updated the Supplementary Material to add the following experiments and analysis:

- Based on comments from Reviewer 3F2p and iHbx, we added the comparison with EvoPrompt [1], a recent evolution-based prompt optimization approach. Please see below for a summary of results, and we added the full discussion in Appendix M.
- Based on comments from Reviewer 3F2p, we added an ablation study on the performance when generating all instructions in a single step without entering into the optimization procedure, and the results show that our iterative procedure performs much better. We summarized the results as the “Comparison with one-step instruction generation” in Appendix K.
- Based on comments from Reviewer 3F2p, we added an ablation study on the number of past solutions to show in the meta-prompt. We summarized the results as the last paragraph of Appendix K.
- Based on comments from Reviewer 3F2p, we added the overfitting analysis in Appendix L. Our results show that when setting aside a validation set for prompt optimization, the validation accuracy trends up and down alongside the training accuracy, hence the training accuracy itself is indicative of test performance.

**Empirical comparison to EvoPrompt**

We added an empirical comparison to EvoPrompt, which is an evolutionary method pointed out by Reviewer iHbx and was also requested by Reviewer 3F2p. At each step, EvoPrompt crosses over and mutates 2 previous prompts to generate a new one. We presented the results and full discussion in Appendix M of the Supplementary Material, and we summarized the key findings as follows:

- On GSM8K, when starting from “Let's solve the problem.” and “Here is the answer.” as the initial instructions, OPRO steadily improves the performance with more steps, while EvoPrompt even degrades the performance compared to the initial prompts (the highest final training accuracies at the 150th step: 75.5 for OPRO, 29.1 for EvoPrompt GA, 25.3 for EvoPrompt DE).
- On BBH sports_understanding, we evaluated a simpler setting where we provided all approaches with task-specific initial instructions, i.e., “Solve the sports understanding problem.” and “Give me the answer to sports understanding.”. In this case, the best version of EvoPrompt improved the initial prompts, but the curve is less stable than OPRO (the highest final training accuracies at the 200th step: 88.0 for OPRO, 62.0 for EvoPrompt GA, 88.0 for EvoPrompt DE).

We believe the following differences between OPRO and evolutionary methods like EvoPrompt are the major reasons why OPRO achieved better results:

- OPRO provides the LLM with richer information to facilitate the understanding of the optimization problem, including task exemplars and accuracies of different prompts. Therefore, OPRO can optimize simple and generic prompts, and can even start from empty strings as shown in our main experiments. On the other hand, evolutionary methods generate new prompts purely based on two past prompts, which is insufficient for understanding the task.
- OPRO utilizes the optimization trajectory of past prompts along with their accuracies, which provides the LLM with more comprehensive context to identify common characteristics of better and worse prompts. On the contrary, 2 prompts alone can be insufficient to infer promising directions to improve the performance, resulting in unstable accuracy curves.
We will respond to each reviewer’s other concerns and questions in the individual responses. Please do not hesitate to ask for any further clarifications.


References:

[1] Qingyan Guo, Rui Wang, Junliang Guo, Bei Li, Kaitao Song, Xu Tan, Guoqing Liu, Jiang Bian, Yujiu Yang. Connecting Large Language Models with Evolutionary Algorithms Yields Powerful Prompt Optimizers. arXiv:2309.08532.

---

### Meta-Review · Area_Chair_8hTx · 2023-12-06

**Metareview:**

This paper investigates the use of Large Language Models (LLMs) as optimizers. The proposed approach prompts an LLM to output a next candidate input, given a sequence of (input, score) pairs as well as some (meta-)instruction in the prompt. The paper demonstrates the performance of LLMs as black-box optimizers on the prompt optimization task (mainly) and also on classical optimization problems such as traveling salesman problem and linear regression.

Overall, the reviewers find the idea investigated in the paper to be interesting, and the experiments (in particular the prompt optimization ones) well executed. The idea of using LLMs to do optimization could inspire future work. Therefore, I recommend acceptance.

As suggested by several reviewers, I encourage the authors to revise some of the claims to better situate this work, as "LLMs as optimizers" could be a big claim. In particular, the authors may want to emphasize that the paper is mainly about prompt optimization, and also give a pointer to the "Comparison with EvoPrompt" section in the main text. It may also be good to mention the linear regression results in the main text as it may be of interest to the community, even though for this paper they are understandably preliminary and more like a motivating example for the prompt optimization task considered later.

**Justification For Why Not Higher Score:**

The obtained results are mainly for prompt optimization (with preliminary results on traveling salesman problem and linear regression), and do not yet cover a broader set of optimization tasks.

**Justification For Why Not Lower Score:**

The idea of using LLMs as optimizers is interesting and could motivate many future studies.

---

### Decision · Program_Chairs · 2024-01-16

Accept (poster)